# Microglial metabolic flexibility supports immune surveillance of the brain parenchyma

Louis-Philippe Bernier [1,2✉], Elisa M. York [1,2], Alireza Kamyabi [1], Hyun B. Choi [1], Nicholas L. Weilinger [1] & Brian A. MacVicar [1✉]

Microglia are highly motile cells that continuously monitor the brain environment and respond to damage-associated cues. While glucose is the main energy substrate used by neurons in the brain, the nutrients metabolized by microglia to support surveillance of the parenchyma remain unexplored. Here, we use fluorescence lifetime imaging of intracellular NAD(P)H and time-lapse two-photon imaging of microglial dynamics in vivo and in situ, to show unique aspects of the microglial metabolic signature in the brain. Microglia are metabolically flexible and can rapidly adapt to consume glutamine as an alternative metabolic fuel in the absence of glucose. During insulin-induced hypoglycemia in vivo or in aglycemia in acute brain slices, glutaminolysis supports the maintenance of microglial process motility and damage-sensing functions. This metabolic shift sustains mitochondrial metabolism and requires mTOR-dependent signaling. This remarkable plasticity allows microglia to maintain their critical surveillance and phagocytic roles, even after brain neuroenergetic homeostasis is compromised.

[1] University of British Columbia, Djavad Mowafaghian Centre for Brain Health, Vancouver, British Columbia, Canada. [2] These authors contributed equally: Louis-Philippe Bernier, Elisa M. York. ✉email: lp.bernier@ubc.ca; bmacvicar@brain.ubc.ca

Microglia in the adult brain under resting conditions possess highly ramified motile processes that constantly extend and retract to survey the local environment[1–4]. This baseline surveillance behavior is thought to be driven by a balance of extracellular signals[5–10] and microglia-intrinsic pathways[11–13], however the metabolic requirements of this continuous cellular motility has not been investigated. Microglia are the main immune effector cells in the central nervous system (CNS), and like peripheral immune cells, they are the first responders upon pathological changes to homeostasis[3,14–17]. Like macrophages, T cells, and B cells of the periphery, microglia are required to survey their environment, phagocytose debris and release cytokines even in suboptimal energetic conditions. Recent findings in immunometabolism have demonstrated that peripheral immune cells can adapt to varying environmental challenges by metabolizing alternative nutrients other than glucose, such as amino acids or fatty acids[18–25]. This metabolic flexibility allows peripheral immune cells to perform wide-ranging functions in perturbed environments where access to carbon sources vary. While the roles of microglia in immune surveillance are similar to their peripheral counterparts, little is known about the energetic requirements of microglia in the unique metabolic environment of the brain.

Glucose is considered to be the main source of metabolic fuel for the brain, with small stores of glycogen in astrocytes to temporarily (<30 min) support neuronal function during metabolic stress[26–30]. The reliance on continuous glucose supply to the brain becomes apparent when glucose levels are compromised, such as in stroke[31,32], in cases of severe hypoglycemia as seen in diabetic management with insulin[33,34], and during hypoglycorrhachia (low cerebrospinal fluid (CSF) glucose) in patients with bacterial meningitis[35] or GLUT1 deficiency[36]. As neuronal health deteriorates in the absence of glucose, it is critical that microglia perform immune protection and debris clearance functions in energy-deprived environments. Microglia likely play a key role in minimizing the extent of tissue damage soon after metabolic insults occur or later when migrating into ischemic areas.

In addition to glucose, glutamine represents another potential metabolic source found in high concentrations within the brain. Glutamate, the deaminated form of glutamine, is the primary excitatory neurotransmitter in the CNS. In the glutamate-glutamine cycle, synaptically-released glutamate is rapidly transported into astrocytes via GLT-1, where it is converted into glutamine and released back into the extracellular space for neuronal uptake and re-processing to glutamate[37,38]. Glutamine may also be taken up and processed through the metabolic pathway of glutaminolysis, in which it is converted into glutamate by glutaminase (GLS), and further processed by glutamate dehydrogenase (GDH) into α-ketoglutarate for direct entry into the tricarboxylic acid (TCA) cycle to support mitochondrial metabolism[39]. Metabolic adaptation leading to reliance on glutamine occurs most notably in activated T cells[21,22] and glutamine-addicted cancer cells[40,41]. In glutaminolytic cells, further metabolic changes come from the glutamine-dependent regulation of mammalian target of rapamycin (mTOR), a serine/threonine kinase that acts as a master regulator of cellular growth and metabolism by regulating a variety of processes, from autophagy to glycolysis[42–44].

In this study, we aimed to investigate the metabolic profile of microglia, assess their ability to function in energy-deficient environments, and explore their potential metabolic flexibility. We demonstrate in vivo, in situ and in vitro that microglia, although highly glycolytic under resting conditions, adapt to glucose deprivation by rapidly switching their energy use to glutaminolysis in an mTOR-dependent manner. This metabolic reprogramming allows microglia to maintain their critical immune surveillance functions even in glucose-deprived conditions where neuronal function is impaired.

## Results

**Microglial metabolic signature.** Microglia constantly monitor the brain environment under resting conditions. To assess the metabolic requirements of this highly dynamic immune surveillance in situ, we first compared the metabolic profile of microglia to that of surrounding cells of the brain parenchyma. Using fluorescence lifetime imaging (FLIM), we took advantage of the endogenous fluorescence of nicotinamide adenine dinucleotide (NADH), a cofactor critical in cellular metabolism. NAD$^+$ is reduced during glycolysis and TCA cycle metabolism to NADH, which is autofluorescent by two-photon excitation (2P $\lambda_{ex}$: 750 nm: $\lambda_{em}$: 460 nm). In its unbound state (free NADH), NADH has a short fluorescence lifetime of ~400 ps, however upon enzymatic binding (e.g. to complex I of the mitochondrial electron transport chain (ETC)), its lifetime is increased to ~2000 ps (Fig. 1a). Therefore, by measuring the lifetime of NADH, it is possible to estimate the ratio of glycolysis to mitochondrial oxidative phosphorylation (OXPHOS)[45,46] and to monitor relative changes in the ratio during metabolic challenges.

Multiple studies have reported a decrease in mean NADH lifetimes when the ETC is inhibited[47,48], suggesting that the mitochondrial NADH signal is mainly correlated with Complex I binding and OXPHOS activity[49,50], although additional minor binding partners exist, such as lactate dehydrogenase and malate dehydrogenase. Following ETC inhibition with antimycin A treatment (Complex III inhibitor, Supplementary Fig. 1), we observed a decrease in the mean lifetime of microglia, supporting the model that bound, long lifetime NADH reflects ETC activity. In addition, glycolytic inhibition with iodoacetate (GAPDH inhibitor, applied along with pyruvate to maintain TCA cycle and ETC activity (Supplementary Fig. 1) induced an increase in mean lifetime in microglia, confirming that glycolytic activity correlates with measurements of free, short lifetime NADH.

Furthermore, since NADH fluorescence cannot be experimentally distinguished from the closely related species NADPH[51], we hereafter refer to our measurements as NAD(P)H. However, NADPH likely represents a negligible number of photons in our NAD(P)H measurements, as previous studies quantifying pyridine dinucleotides in the brain report NADH concentrations to be approximately 5 to 10 fold higher than NADPH[52,53]. Accordingly, our readings of NAD(P)H lifetimes in response to metabolic manipulations (iodoacetate or antimycin A, Supplementary Fig. 1) perform as expected from the NADH species. Thus, our data can be interpreted within the simplified schematic where short NAD(P)H lifetime correlates with glycolysis and longer NAD(P)H lifetime with OXPHOS activity.

To identify microglia in situ, we used acute brain slices from wild type mice and labeled microglia with DyLight 594 tomato lectin. This avoids any artefactual interference of NAD(P)H fluorescence signal by EGFP, which we recently showed occurs in the commonly used CX3CR1$^{EGFP/+}$ microglial reporter mouse[54]. Using this approach, we observed that microglia display a much shorter mean NAD(P)H lifetime than the surrounding neuropil (non-microglial cells). This indicates a higher free:bound NAD(P)H ratio, suggesting that under resting conditions microglia have a more glycolytic profile than other cells of the neuropil (Fig. 1b–d).

**Microglial surveillance during hypoglycemia in vivo.** NAD(P)H measurements show that microglia are highly glycolytic relative to the surrounding neuropil. We therefore tested how microglial surveillance of the brain parenchyma is affected by a glucose

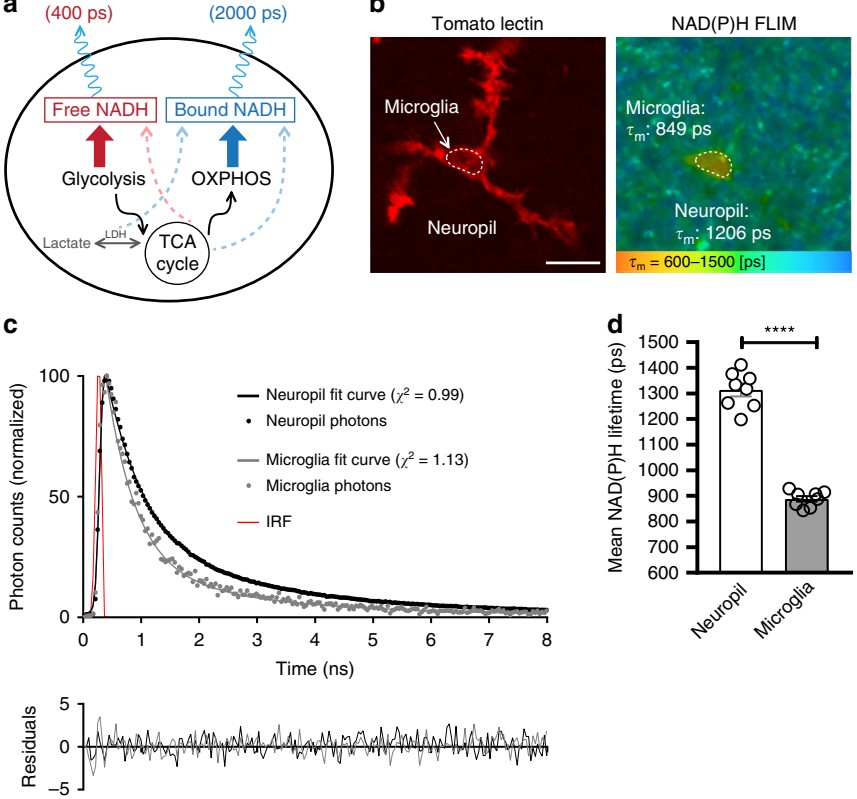

**Fig. 1 Microglial metabolic signature in situ. a** Free NADH mainly produced by glycolysis and TCA cycle has a short fluorescent lifetime of ~400 ps, while NADH bound to proteins, such as electron transport chain components linked to OXPHOS activity, has a longer lifetime of ~2000 ps. Dashed lines indicate other possible sources of free or bound NADH. **b** In acute brain slices, using fluorescence lifetime imaging (FLIM) of NAD(P)H fluorescence, microglia labeled with DyLight 594 tomato lectin show a shorter NAD(P)H lifetime relative to the neuropil. Indicated on the right panel are mean lifetimes of microglial region of interest (ROI) and neuropil ROI. **c** Two-exponential decay curves interpolated from photons from microglia, or surrounding neuropil (shown is total ROI photon count normalized to the peak of the curves, from ROIs shown in **b**. IRF: instrument response function). **d** Quantification of mean NAD(P)H lifetime for neuropil vs microglia (neuropil: 1314 ± 25 ps, microglia: 889 ± 11 ps; ****$p = 0.0000000003$, unpaired, two-tailed $t$-test; $n = 8$ mice, data are represented as mean ± SEM). Scale bar, 10 µm (**b**). Source data are provided as a Source Data file. See also Supplementary Fig. 1.

decrease in an insulin-induced hypoglycemia in vivo model. Microglia reporter transgenic mice (CX3CR1[EGFP/+]) were injected with 5 U/kg insulin, and the subsequent drop in blood glucose levels was monitored along with microglial dynamics (Fig. 2a). Moderate to severe hypoglycemia was achieved 45–60 min following injection, with blood and CSF glucose concentration decreasing to an average of 3.27 mM and 3.21 mM, respectively (compared with 22.55 mM and 7.19 mM 45–60 min after vehicle control injection; Fig. 2b). Following at least 30 min of moderate to severe hypoglycemia (blood [Glc] < 5 mM), mice were fixed and the ramified morphologies of hippocampal microglia were investigated (average blood [Glc] at time of fixation = 3.03 mM; Fig. 2c). Surprisingly, microglia in hypoglycemic mice showed no apparent reduction in their ramified morphology when compared to sham-injected animals, even showing a slight increase in the number of branch points per microglia when quantified using 3DMorph[55] (Fig. 2d–g). The effect of hypoglycemia on microglial process motility was also assessed. In vivo two-photon time-lapse imaging of cortical microglia through a cranial window showed no effect of sustained hypoglycemia on the motile surveillance of the brain parenchyma by microglia (Fig. 2h, i, Supplementary Movie 1). The damage-sensing response of microglia was similarly unaffected by severe hypoglycemic conditions (blood [Glc] <2.5 mM). 75 min after insulin injection, microglia showed robust responses to a laser-induced lesion, extending their large processes to converge on the damaged area (Fig. 2j–l, Supplementary Movie 1). Taken together, our data suggests that microglial

function and motility in the brain is unaffected by prolonged (up to 90 min) reductions in extracellular glucose levels.

**Microglial surveillance and metabolism during aglycemia.** To further investigate the possibility of flexible nutrient use in microglia, we imaged acute hippocampal slices of CX3CR1[EGFP/+] mice during complete aglycemia (0 mM glucose in artificial CSF (aCSF)—absence of glucose in the tissue confirmed experimentally in Supplementary Fig. 2. No exogenous carbon sources were added). Following 60 min of aglycemia, the microglial morphology was unaltered compared to control incubation (Fig. 3a, b). Using 3DMorph, we quantified microglial ramification, number of branch points, as well as the percentage of parenchymal territorial volume occupied by microglia and found no significant alterations of microglial morphology after sustained absence of glucose (Fig. 3c–g). Two-photon time-lapse imaging showed that the continuous process motility that characterizes microglial surveillance of the brain parenchyma continued unaltered during 1 h of aglycemia (Fig. 3h, i, Supplementary Movie 2). A small but significant increase in motility index was even observed following 1.5 h of glucose-free aCSF when compared to a similar incubation in control aCSF (Fig. 3j), possibly suggesting an active response to the altered extracellular milieu in these conditions where neuronal function is lowered[56]. Following 1 to 1.5 h of aglycemia, microglia also rapidly extended their processes to converge towards local laser-induced damage in a manner indistinguishable from control-treated slices (Fig. 3k–n, Supplementary Movies 3, 4).

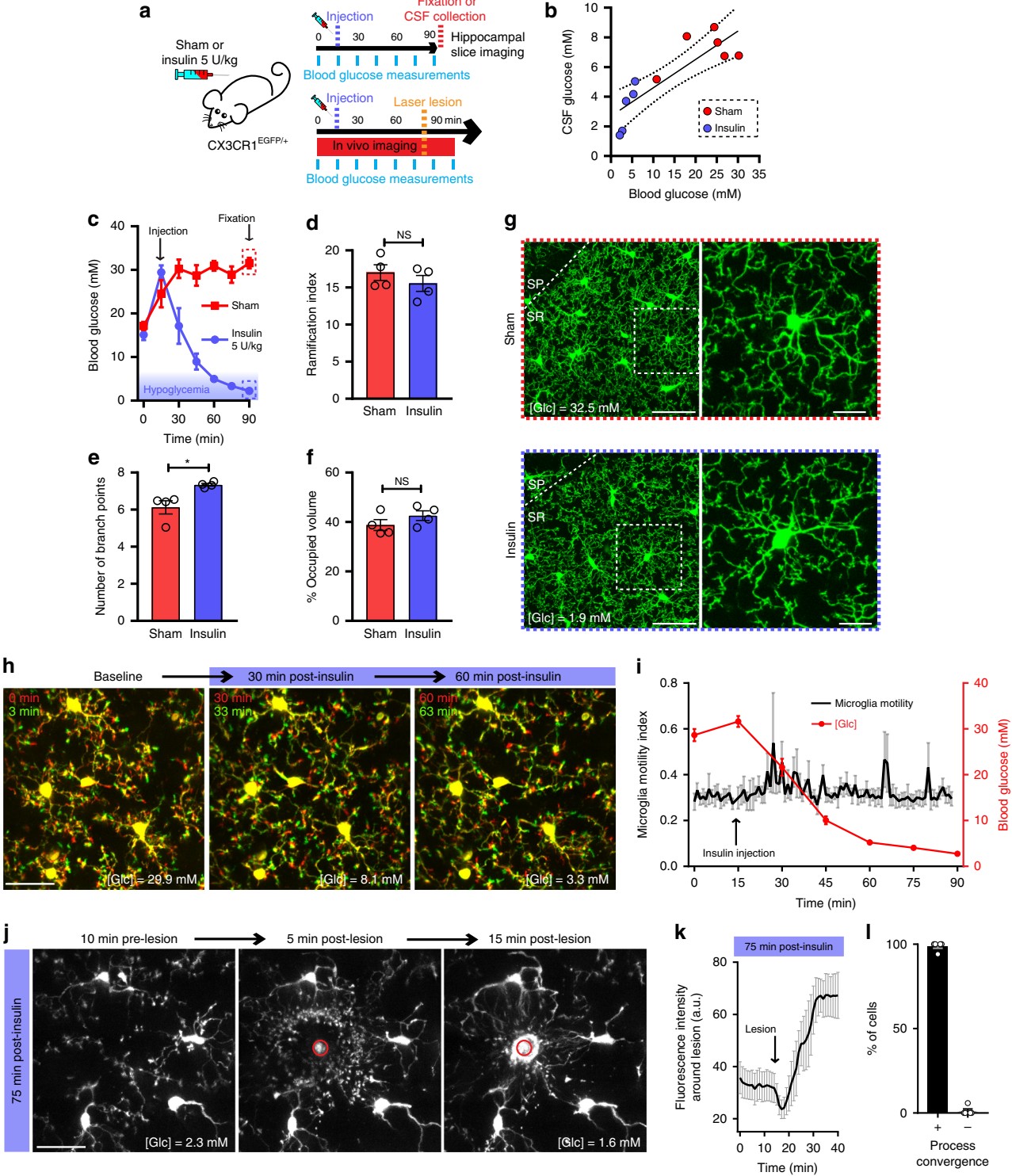

Since microglia showed no apparent change in their motile behavior in the prolonged and complete absence of glucose, we investigated the possibility that microglia can rapidly reprogram their metabolism and transform their intracellular metabolic signature. We imaged NAD(P)H lifetime in brain slices incubated in glucose-containing or aglycemic aCSF for one hour (Fig. 4a–c). In the neuropil, we observed a time-dependent decrease in NAD(P)H lifetime that was similar in both control aCSF and in aglycemia (Fig. 4b–d, Supplementary Fig. 3). On the other hand, microglial NAD(P)H lifetime

remained constant over time in normoglycemia and, more importantly, was significantly increased in response to the removal of glucose (Fig. 4b–f, Supplementary Fig. 3). This rise in NAD(P)H lifetime stems from a relative increase in the bound:free NAD(P)H ratio, indicating that microglia adapt to aglycemia by increasing their relative rate of mitochondrial metabolism. Therefore, microglia show a unique ability to sustain oxidative phosphorylation in the absence of glucose, suggesting that they are metabolizing an alternate carbon source to supply TCA metabolites and maintain OXPHOS.

**Fig. 2 Microglia maintain their surveillance function during hypoglycemia in vivo. a** CX3CR1$^{EGFP/+}$ mice were injected with saline (sham) or insulin (5 U/kg). Blood glucose was measured every 15 min. For panel **a–g**, either CSF collection or PFA fixation for imaging was done 75 min after injection (SR: stratum radiatum, SP: stratum pyramidale). For panel **h–l**, time-lapse imaging was performed through a cranial window. **b** Correlation between CSF glucose concentration and blood glucose concentration (linear regression $R^2 = 0.724$, 95% confidence interval shown as dashed lines, $n = 5$ insulin-injected mice, 6 sham-injected mice). **c** Blood glucose concentration over time following insulin or sham injection ($n = 4$ mice per treatment. Note that as previously reported[110], anesthesia induced an increase in blood glucose levels). **d** Ramification index of microglia 75 min after insulin or sham injection (for panels **d–g**, sham: $n = 4$ mice, 114 cells; insulin: $n = 4$ mice, 143 cells). **e** Number of branch points per microglia (*$p = 0.0162$, unpaired, two-tailed $t$-test). **f** Percentage of brain volume occupied by microglia. **g** Hippocampal microglial morphology 75 min after insulin or sham injection. [Glc] indicated is the last blood glucose measurement obtained prior to fixation. **h** Process motility of cortical microglia imaged through a cranial window at indicated time-points following insulin injection. Shown are overlaid images obtained 3 min apart to show dynamic microglial motility. [Glc] indicated on individual panels is the blood glucose concentration obtained at that time-point (for **h** and **i**, $n = 5$ mice). **i** Quantification of microglial motility index over time following insulin injection (left axis) as well as blood glucose concentration (right axis) obtained from the same mice. **j** In vivo microglial response to a focal laser-induced lesion 75 to 100 min after insulin injection (for **j** and **k**, $n = 5$ mice). **k** Quantification over time of the change in fluorescence intensity of the area around the lesion (not including the lesion itself) when performed 75 min after insulin injection. **l** 75 min after insulin injection, percentage of microglia surrounding the lesion (within a 75 μm radius of the lesion) that respond (+) or not (−) by converging their processes towards the lesion ($n = 5$ mice). For **d** and **f**, NS, $p > 0.05$, unpaired, two-tailed $t$-test; data are represented as mean ± SEM. Scale bars, 50 μm (**g**, left panels); 25 μm (**g** right panels, **h**, **j**). Source data are provided as a Source Data file.

**Microglia in vitro metabolize glutamine.** Our data suggest that brain-resident microglia are metabolically flexible, consistent with observations in peripheral immune cells[21,22]. However, the metabolic environment of the brain is unique: one interesting feature is the importance of the glutamate-glutamine cycle, in which astrocytes take up and convert glutamate to generate a relatively constant level of extracellular glutamine. To first isolate the possibility of microglia using glutamine as an energy substrate, we proceeded to in vitro microglial culture models. To assess microglial viability and metabolism while controlling the carbon source availability, we performed MTT assays to evaluate cellular viability as well as mitochondrial activity levels via oxidoreduction of a tetrazolium dye by NADH-dependent enzymes in metabolically active cells. Primary microglia from E18 rats were pre-incubated for 4 h in media containing either glucose + glutamine (10 mM + 4 mM, respectively), glucose-only (10 mM), glutamine-only (4 mM), or media deficient for both carbon sources. MTT metabolism by microglia remained high in conditions that contained either glucose or glutamine, but was significantly decreased in deficient conditions (Fig. 5a), suggesting that microglia are capable of metabolizing glutamine. We additionally addressed the mitochondrial metabolism of microglia exposed to defined glucose and/or glutamine conditions by measuring their oxygen consumption rate (OCR) using the Seahorse Extracellular Flux Analyzer mitochondrial stress test. Primary microglia were plated into the Seahorse reader with glucose +glutamine, glucose-only, glutamine-only or deficient media and OCR was measured for 4 h to obtain carbon source-specific basal mitochondrial respiration rates (Fig. 5b–d, Supplementary Fig. 4). The OCR of microglia was found to be similar in both glucose-only and glutamine-only conditions. This was significantly higher than in deficient media, again indicating microglial OXPHOS can function with either glucose or glutamine supplying the TCA cycle. After the 4-h incubation with specific carbon sources, we ran a mitochondrial stress test, consisting of successive applications of the metabolic inhibitors oligomycin (to inhibit ATP synthase and reveal ATP-linked respiration), carbonyl cyanide-4 (trifluoromethoxy) phenylhydrazone (FCCP; a mitochondrial protonophore driving maximal respiration), and rotenone with antimycin A (inhibitors of complex I and complex III, respectively, to block ETC activity and reveal any remaining non-mitochondrial oxygen consumption). When the uncoupler FCCP is added to dissipate the proton gradient across the mitochondrial membrane, the electron transport chain is uninhibited and maximal oxygen consumption rate is achieved. The maximal mitochondrial respiratory capacity of microglia can therefore be inferred and compared between the 4 different carbon source conditions. The use of glutamine as the only carbon source was as efficient as glucose at sustaining maximal mitochondrial respiration (Fig. 5b, d). Here again, the maximal respiration rate was significantly higher than in deficient media. Taken together, our data shows that microglia can maintain mitochondrial activity using glutamine in the absence of glucose.

**Glutaminolysis supports microglia metabolism under aglycemia.** Microglia can maintain their mitochondrial activity solely using glutamine, suggesting that during aglycemic challenge, glutaminolysis supplies the metabolites for the TCA cycle and supports the production of NADH necessary for OXPHOS. We next sought to determine whether disrupting glutaminolysis inhibited microglial metabolic flexibility in situ. As demonstrated in Fig. 4, one hour of aglycemia treatment induced an adaptive increase in microglial NAD(P)H lifetime, linked to a relative increase in OXPHOS. However, after 60 min in the presence of the glutaminolysis inhibitor epigallocatechin gallate (EGCG) in addition to glucose removal, the NAD(P)H lifetime decreased, showing that glutaminolysis is necessary to maintain mitochondrial function in microglia (Fig. 6a–c). Importantly, EGCG will block the conversion of glutamate to α-ketoglutarate, thereby inhibiting its metabolism without directly preventing the glutamate-glutamine cycle needed for neuronal function in brain tissue. This was confirmed by a similar reduction in NAD(P)H lifetime observed during aglycemia in the presence of R162 (200 μM), another GDH inhibitor (Fig. 6d). In these conditions, a decrease in the NAD(P)H lifetime is expected, as aglycemia without the ability to metabolize alternative carbon substrates and replenish the TCA cycle and ETC metabolism will result in a corresponding decrease of both free and bound NAD(P)H. This indicates that in the absence of glucose, microglia switch to glutaminolysis to support the TCA cycle and mitochondrial respiration. Extracellular concentration of glutamine in the brain is estimated at around 385 μM[57], and we tested its presence in acute brain slices to confirm its suitability as a sufficient nutrient during aglycemia (Supplementary Fig. 2). After one hour of incubation in aCSF, brain slice glutamine content was still 47.5% of its initial pre-incubation level. Some of this decrease may be due to dilution in aCSF, but the rundown is also likely caused by metabolic conversion of glutamine into α-ketoglutarate and therefore out of the glutamate-glutamine cycle. Accordingly, aglycemia induced a larger rundown of glutamine (to 33.8% after a one hour incubation), suggesting more glutamine is being metabolized as an alternate energy substrate.

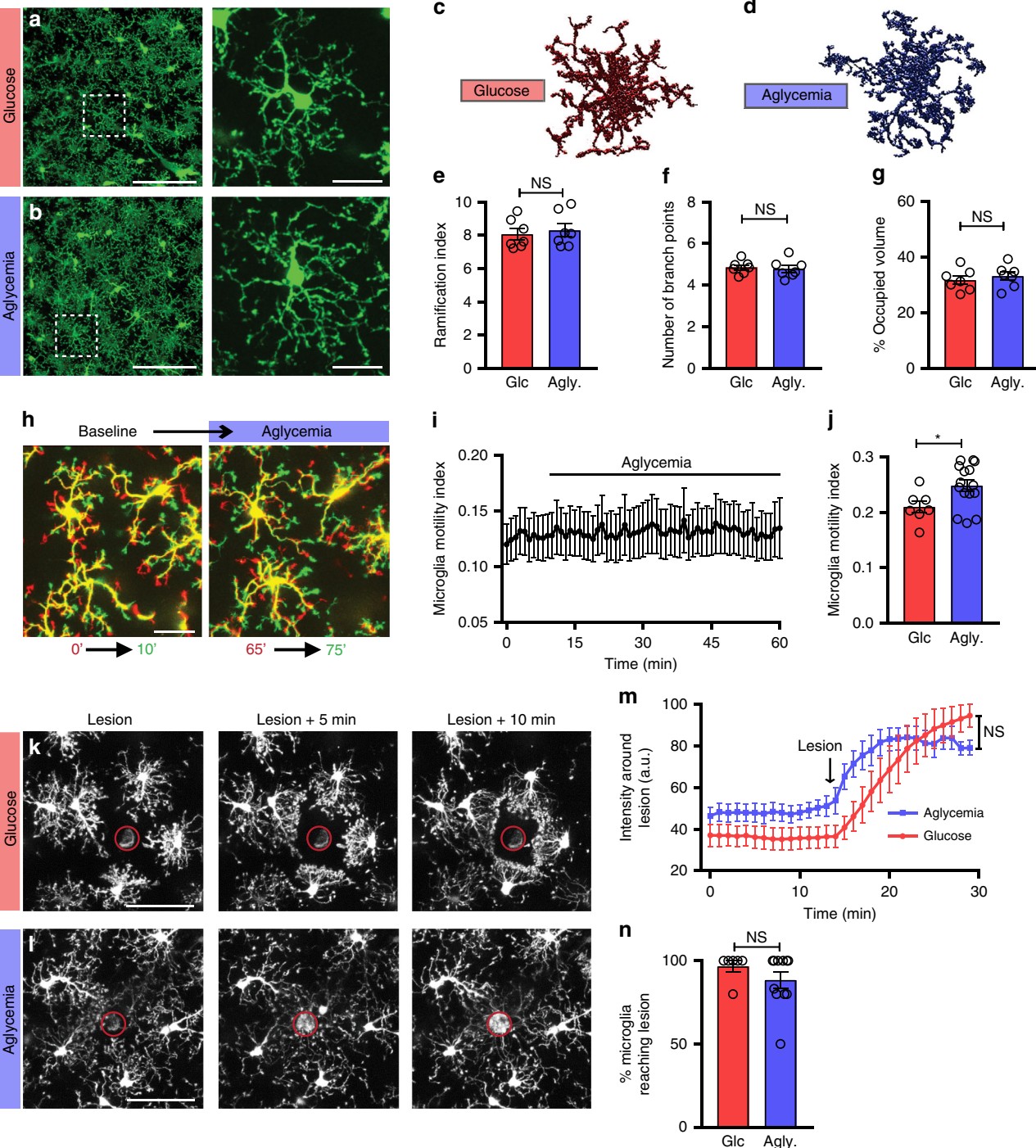

**Fig. 3 Microglial surveillance is unaltered by aglycemia in situ. a, b** Microglial morphology following a 60-min incubation in glucose-containing aCSF (**a**, 10 mM glucose) or aglycemic aCSF (**b**, 0 mM glucose) (for panel **a–g**, glucose aCSF: $n = 7$ mice, 14 slices, 307 cells; aglycemia: $n = 7$ mice, 14 slices, 322 cells). **c, d** 3DMorph rendering of representative microglial cells following a 60-min incubation in glucose-containing aCSF (**c**) or aglycemic aCSF (**d**). **e** Ramification index of microglia 60 min after control or aglycemia incubation. **f** Number of branch points per microglia. **g** Percentage of brain volume occupied by microglia. **h** Process motility of microglia following one hour of aglycemia. Shown are overlaid images obtained 10 min apart to show dynamic microglial motility (for **h–j**, $n = 7$ slices for glucose, 15 slices for aglycemia). **i** Quantification of microglial motility index over time during aglycemia treatment. **j** Microglial motility index after 60-min incubations in glucose-containing aCSF or aglycemic aCSF. Shown here is the mean microglial motility for a 15-min imaging period following the incubation time (*$p = 0.0319$, unpaired, two-tailed $t$-test). **k, l** Microglial response to a laser-induced lesion following a 60-min incubation in glucose-containing aCSF (**k**) or aglycemic aCSF (**l**) (for **k–n**, $n = 5$ slices for glucose, 7 slices for aglycemia). **m** Quantification over time of the change in fluorescence intensity of the area around the lesion (not including the lesion). **n** Percentage of microglia surrounding the lesion (within a 75 μm radius of the lesion) with processes reaching the lesion within 20 min ($n = 6$ slices for glucose, 11 slices for aglycemia). For **e–g, n**, NS, $p > 0.05$, unpaired, two-tailed $t$-test; for **m**, treatment factor is NS, $p > 0.05$, but time factor is ****$p < 0.0001$, two-way ANOVA; data are represented as mean ± SEM. Scale bars, 100 μm (**a, b**, left panels); 20 μm (**a, b**, right panels); 25 μm **h**; 50 μm **k, l**. Source data are provided as a Source Data file. See also Supplementary Fig. 2.

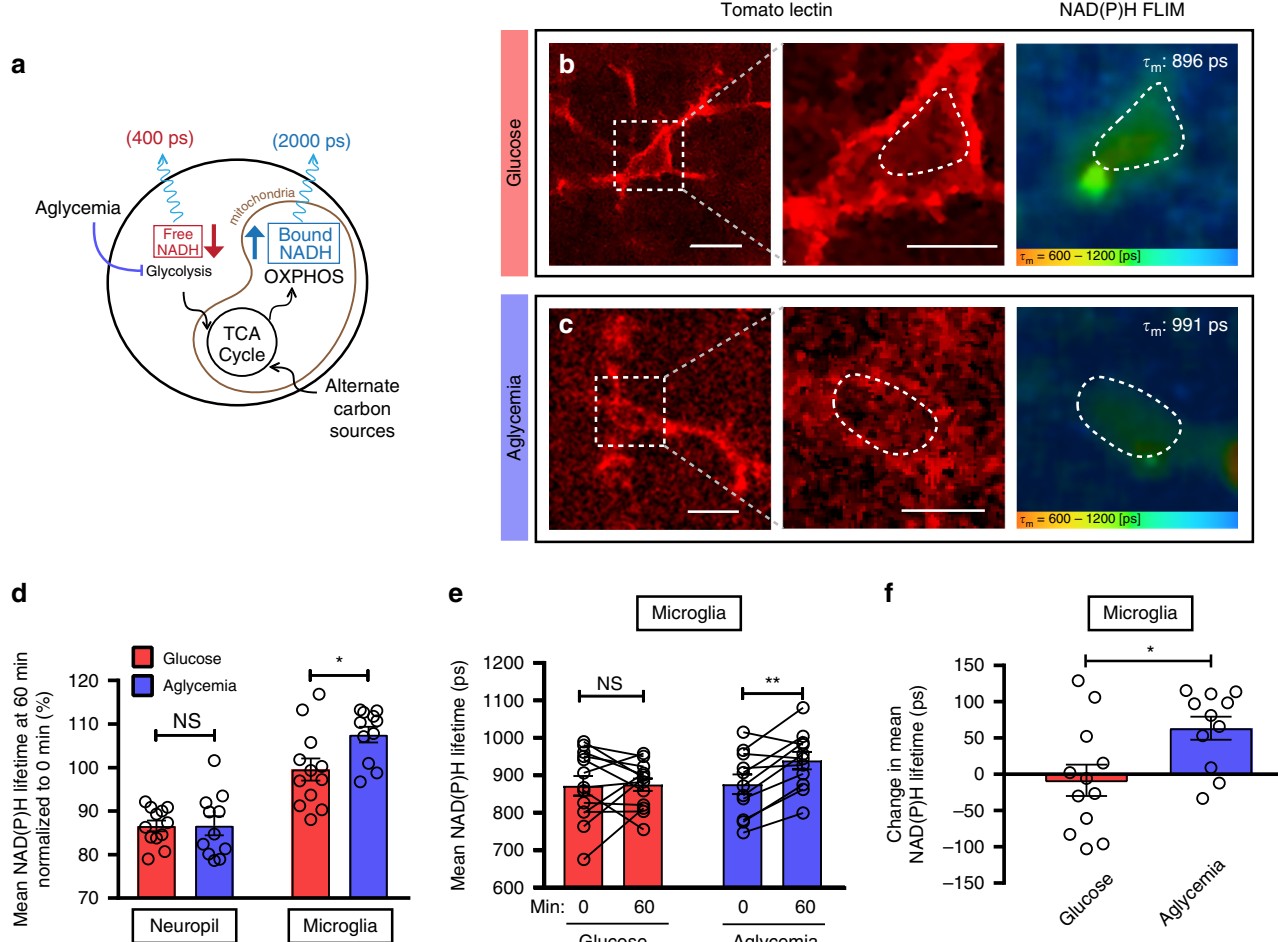

**Fig. 4 Microglial metabolism is unaltered by aglycemia in situ. a** Schematic diagram showing the effect of aglycemia on free and bound NADH levels. **b**, **c** Representative images of FLIM-NAD(P)H imaging of microglia labeled with tomato lectin, following a 60-min incubation in glucose-containing aCSF (**b**) or aglycemic aCSF **c**. Indicated on the right panel is the mean NAD(P)H lifetime of the microglial ROI within the dotted line (for **b–f**, $n = 4$ mice, 12 microglia and surrounding neuropil for control, $n = 4$ mice, 11 microglia and surrounding neuropil for aglycemia). **d** Mean NAD(P)H lifetime of neuropil vs microglia, obtained 60 min after incubation in glucose-containing aCSF or aglycemic aCSF. Values are normalized to their respective baseline from $t = 0$. (NS, $p > 0.05$; *$p = 0.0204$, unpaired, two-tailed $t$-test). **e** Mean NAD(P)H lifetime of microglia from 0 to 60 min when incubated in either glucose-containing aCSF or aglycemic aCSF (NS, $p > 0.05$; **$p = 0.0027$, paired, two-tailed $t$-test). **f** Relative change in mean NAD(P)H lifetime of microglia over 60 min of treatment (*$p = 0.0158$, unpaired, two-tailed $t$-test). Data are represented as mean ± SEM. Scale bars, 10 μm (**b**, **c**, left panels); 5 μm (**b**, **c**, middle panels). Source data are provided as a Source Data file. See also Supplementary Fig. 3.

**Glutaminolysis supports microglial surveillance in aglycemia.** We then tested whether glutamine is also required for the maintained immune surveillance we observed during prolonged glucose reductions in vivo and in situ. We subjected brain slices from CX3CR1$^{EGFP/+}$ mice to one hour of complete aglycemia without addition of an exogenous carbon source, and microglia still showed typical ramified morphology as quantified in Fig. 3 (Fig. 7a). However, inhibiting glutaminolysis by blocking the key enzyme GDH with EGCG during 60 min of aglycemia caused a visible microglial deramification, where cell bodies became enlarged with short and stubby protrusions (Fig. 7b). Microglial morphology was quantified and significant reductions in ramification index, number of branch points, and territorial volume occupied by microglia were found in aglycemia when combined with glutaminolysis inhibition compared to aglycemia alone (Fig. 7c–g). The microglial baseline motility and sensitivity to neighboring damage was also assessed following 1- to 1.5-h aglycemia treatment, with or without EGCG (Fig. 7h–l). Glutaminolysis inhibition in the absence of glucose significantly reduced the baseline motility of microglial large processes (Fig. 7j, Supplementary Movie 5). This also led to the microglial damage response being hampered, with few

microglia extending their processes towards local laser-induced lesions (Fig. 7h, i, k, l, Supplementary Movie 6). This shows that microglia are flexible in their metabolic fuel usage with glutaminolysis providing alternative energy support for essential microglial surveillance activity.

Furthermore, we examined whether glutamine is also required by microglia for metabolism and immune surveillance function in the continued presence of glucose. When both carbon sources are unaltered in the environment, blocking glutaminolysis still induced morphological changes in microglia in situ after one hour incubation (Supplementary Fig. 5). NAD(P)H lifetime in microglia was reduced by EGCG suggesting a possible role for glutaminolysis in microglial metabolism in the presence of glucose (Supplementary Fig. 5). In our in vitro Seahorse assay, the removal of glutamine (i.e., in glucose-only media) had significant effects on basal OCR and maximal respiration (Fig. 5b–d), again indicating that glutamine may be required for microglial function, even when glucose is available.

**Fatty acid oxidation involvement in microglial functions.** While our results show that glutamine is the main alternate

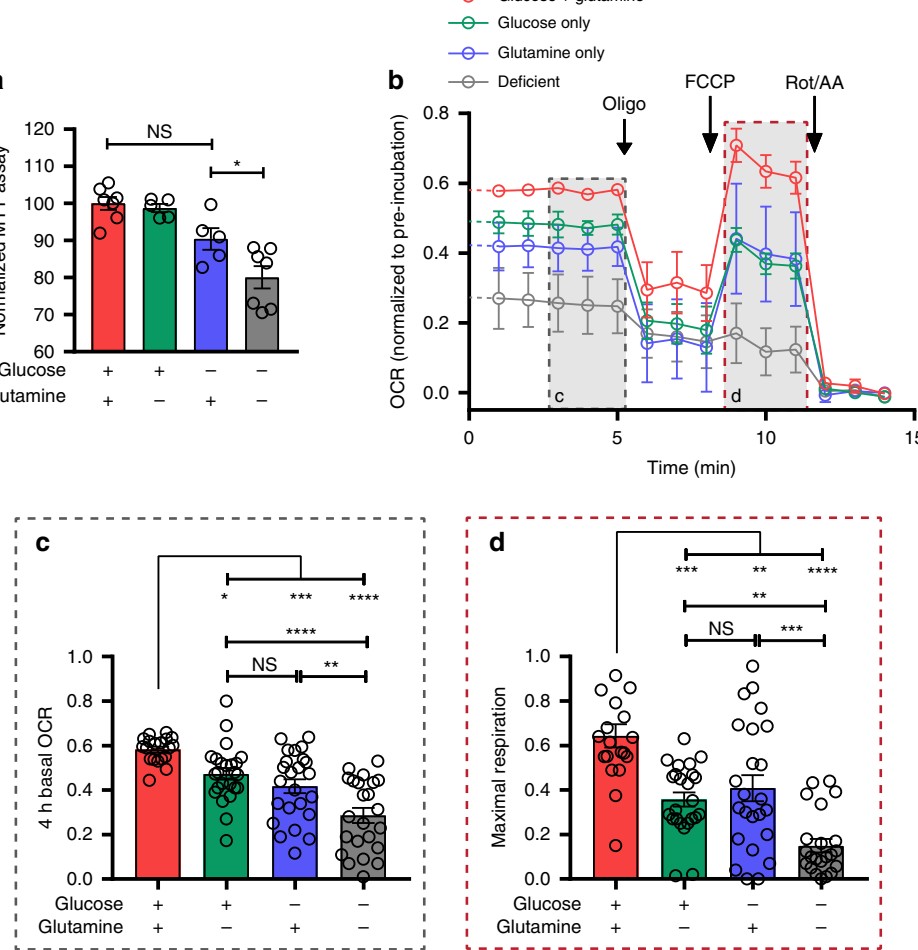

**Fig. 5 Microglia metabolize glutamine in vitro. a** MTT oxidoreduction by primary microglia following a 4-h incubation in defined carbon source media. Values were normalized to that of control (glucose+glutamine) condition within the respective individual experimental trial (glucose+glutamine: $n = 7$; glucose: $n = 5$; glutamine: $n = 5$; deficient: $n = 7$. $n$ Values represent wells, from 3 experimental trials; *$p = 0.0349$; one-way ANOVA). **b** Oxygen consumption rate (OCR) of primary microglia during a Seahorse Extracellular Flux Analyzer mitochondrial stress test following a 4-h pre-treatment in control (glucose+glutamine), glucose-only, glutamine-only or deficient (no glucose, no glutamine) media (for **b–d**, $n = 3$ experimental trials (Seahorse plates); glucose + glutamine: 19 wells; glucose: 24 wells; glutamine: 24 wells; deficient: 23 wells). OCR values are normalized to initial reading at time point 0 for each well. Gray insets show the metabolic parameters quantified in **c** and **d**. **c** Basal OCR of microglia after 4-h pre-treatment in defined carbon source media (NS, $p > 0.05$; *$p = 0.0464$; **$p = 0.0061$; ***$p = 0.0008$; glucose+glutamine vs deficient: ****$p = 0.000000002$; glucose vs deficient: ****$p = 0.00004$; one-way ANOVA). **d** Maximal mitochondrial respiration of microglia, following FCCP addition after 4-h pre-treatment in defined carbon source media (NS, $p > 0.05$; glucose+glutamine vs glutamine: **$p = 0.0027$; glucose vs deficient: **$p = 0.0062$; glucose+glutamine vs glucose: ***$p = 0.0002$; glutamine vs deficient: ***$p = 0.0004$; ****$p = 0.0000000005$; one-way ANOVA). Data are represented as mean ± SEM. Source data are provided as a Source Data file. See also Supplementary Fig. 4.

carbon source for microglia, we also explored the possible involvement of fatty acid β-oxidation in fueling glial functions during metabolic challenges. Fatty acid oxidation (FAO) is believed to be seldom used in brain cell metabolism, notably due to poor transport across the blood-brain barrier, slow ATP generation, and superoxide generation potential[58–60]. However, previous studies as well as expression and transcription patterns of FAO enzymes suggest that glial cells, but not neurons, may exploit fatty acids for energy[61–66]. The rate-limiting step in the enzymatic metabolism of fatty acids is the conversion of fatty acyl-CoA to acyl-carnitines for translocation into the mitochondria. This is catalyzed by carnitine palmitoyltransferase 1 (CPT1), an enzyme that can be experimentally inhibited to evaluate metabolic reliance on fatty acids. We therefore tested the effect of etomoxir (100 μM), a widely used CPT1 inhibitor, on microglial function in situ. In the absence of glucose, FAO inhibition caused microglial deramification and significantly reduced the motile

surveillance behavior and damage-sensing capacities of microglia (Fig. 8a–l). We further confirmed the morphological effects of FAO inhibition with perhexiline, another CPT1 blocker (Supplementary Fig. 6). Fatty acid metabolism in the brain is likely minimal under normoglycemic condition but could be increased under low energy states. We show that brain slices challenged with one hour of aglycemia increase their FAO enzymatic activity by 154.3% (octanoyl-CoA oxidation; Fig. 8m). This whole brain slice approach does not allow identification of the cell type contributing to the FAO activity increase, however, recent studies have suggested that the use of fatty acids for feeding of the TCA cycle in the brain occurs exclusively in astrocytes. The expression of key FAO enzymes CPT1a and CPT2 is observed only in astrocytes[62], and this is corroborated by the analysis of *Cpt1a* transcription pattern from cell type-specific transcriptome databases, further showing that astrocytes are more likely to metabolize fatty acids than microglia (Supplementary Fig. 7). Since the

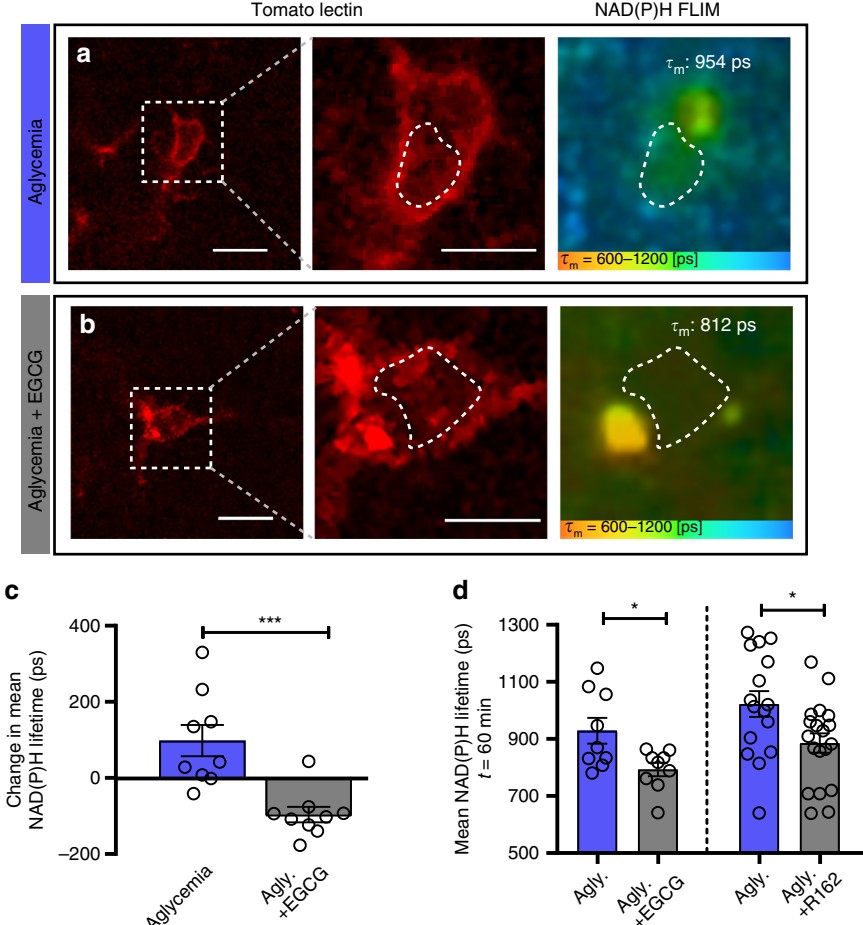

**Fig. 6 Glutaminolysis supports the microglial metabolic adaptation to aglycemia in situ. a, b** Representative images of FLIM-NAD(P)H imaging of microglia labeled with tomato lectin, following a 60-min incubation in aglycemic aCSF (**a**) or aglycemic aCSF with EGCG (**b**). Indicated on the right panel is the mean NAD(P)H lifetime of the microglial ROI within the dotted line (for **a**–**d**: $n = 3$ mice, 9 microglia for aglycemia, $n = 3$ mice, 9 microglia for aglycemia+EGCG). **c** Relative change in mean NAD(P)H lifetime of microglia over 60 min of treatment (***$p = 0.0006$, unpaired, two-tailed $t$-test). **d** Left bars, mean NAD(P)H lifetime of microglia after a 60-min incubation in aglycemia aCSF, with or without EGCG (100 μM; *$p = 0.0174$, unpaired, two-tailed $t$-test). Right bars, mean NAD(P)H lifetime of microglia after a 60-min incubation in aglycemia aCSF, with or without R162 (200 μM; *$p = 0.0200$, unpaired, two-tailed $t$-test; $n = 3$ mice, 16 microglia for aglycemia, $n = 3$ mice, 19 microglia for aglycemia+R162). Data are represented as mean ± SEM. Source data are provided as a Source Data file. See also Supplementary Fig. 5.

conversion of glutamate to glutamine occurs exclusively in astrocytes, we hypothesize that astrocytic CPT1 inhibition during aglycemia hinders the ability of astrocytes to produce glutamine necessary for microglial metabolism, thereby indirectly reducing microglial surveillance.

**Microglial metabolic flexibility is mTOR-dependent**. mTOR is a ubiquitous serine/threonine kinase that has been shown to be essential in sensing nutrient availability and controlling the anabolic-catabolic balance, thereby acting as a master regulator of metabolism. mTOR is known to regulate the glycolysis to glutaminolysis switch and, interestingly, is itself regulated by glutamine[67,68]. We therefore sought to find whether the microglial metabolic flexibility uncovered here relies on mTOR-dependent signaling. Looking at microglial morphology in situ, Torin-1 (mTOR inhibitor) accompanied by aglycemia induced significant process retraction and rounding of the cells, as measured by the number of branch points and the volume of brain parenchyma surveyed by microglia (Fig. 9a–g). Process motility and damage-sensing function were also negatively altered by mTOR inhibition during aglycemic challenge (Fig. 9h–l, Supplementary Movie 7). Interestingly, mTOR had no effect on microglial function when

nutrient availability was unperturbed (Supplementary Fig. 8), indicating that mTOR signaling is only essential when a switch in metabolic pathway is required to take place due to environmental pressures. Taken together, our data shows that microglia, when deprived of glucose, rapidly switch to glutamine metabolism in an mTOR-dependent manner to temporarily maintain their brain surveillance function (Fig. 9m).

## Discussion

As the first line of host defense, cells of the immune system need to function in varying environments including those in which nutrient availability is compromised or altered. They perform various cellular tasks and undergo rapid fluctuations between phenotypic states, all of which dramatically affect their energy demands[18,20]. Immune cells have therefore evolved into remarkably adaptable cells. By investigating their metabolic signature in vivo, ex vivo and in vitro, we provide robust evidence that brain-resident microglia rapidly adapt to alternative metabolic sources despite being highly glycolytic in unperturbed conditions. This metabolic flexibility is apparent when environmental glucose is decreased and microglia switch their energy

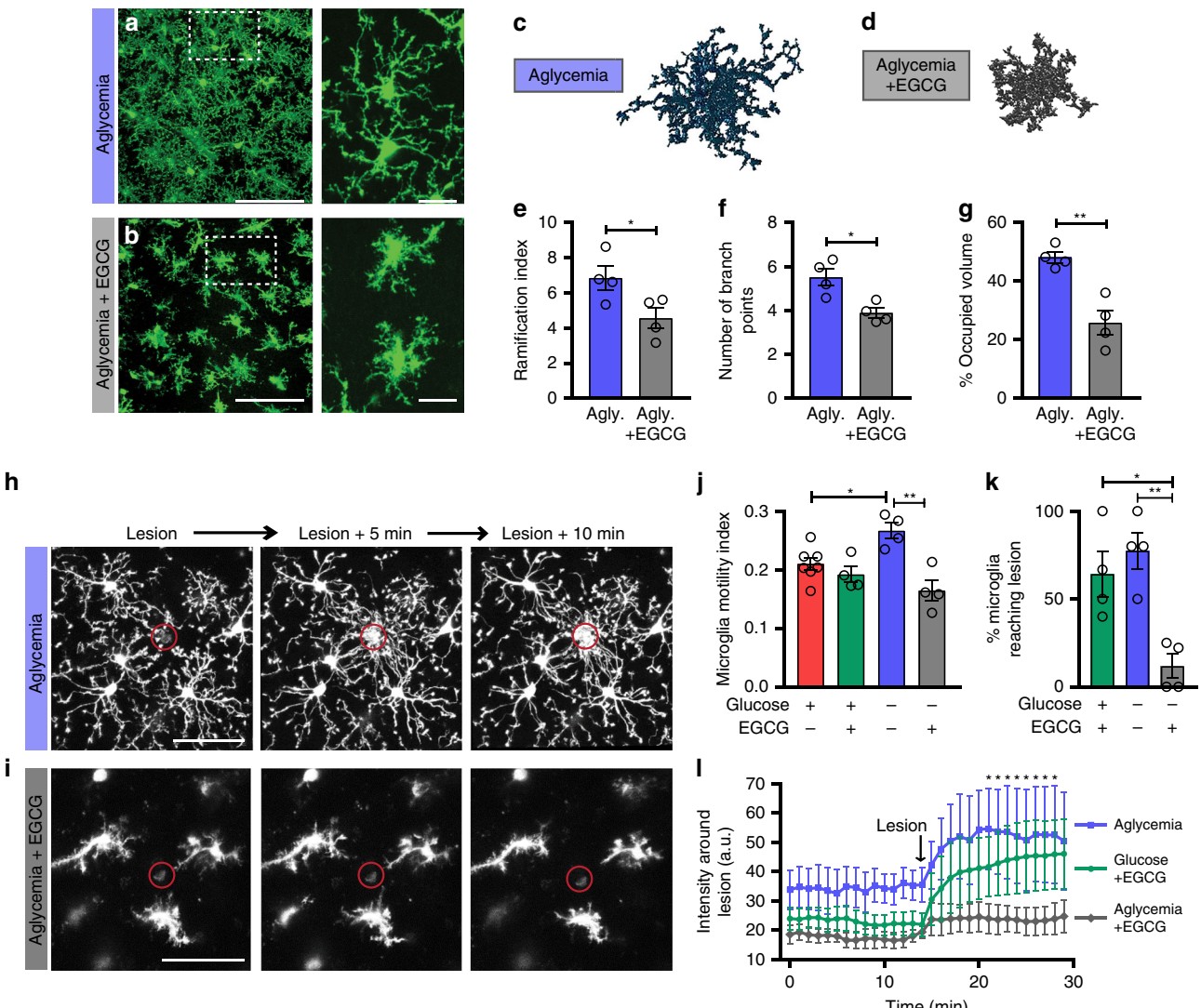

**Fig. 7 Glutaminolysis is required for microglial surveillance during aglycemia in situ. a**, **b** Microglial morphology following a 60-min incubation in aglycemic aCSF **a** or aglycemic aCSF with EGCG **b** (for panel **a–g**, aglycemia: $n = 4$ mice, 8 slices, 139 cells; aglycemia+EGCG: $n = 4$ mice, 8 slices, 173 cells). **c**, **d** 3DMorph rendering of representative microglial cells following a 60-min incubation in aglycemic aCSF **c** or aglycemic aCSF with EGCG **d**. **e** Ramification index of microglia 60 min after aglycemia or aglycemia+EGCG incubation ($*p = 0.0437$, unpaired, two-tailed $t$-test). **f** Number of branch points per microglia ($*p = 0.0111$, unpaired, two-tailed $t$-test). **g** Percentage of brain volume occupied by microglia ($**p = 0.0028$, unpaired, two-tailed $t$-test). **h**, **i** Microglial response to a laser-induced lesion following a 60-min incubation in aglycemic aCSF **h** or aglycemic aCSF with EGCG (**i**) (for **h–l**, $n = 6$ slices for control, 4 slices for treatment conditions). **j** Microglial motility index after 60-min incubations in the indicated conditions. Shown here is the mean microglial motility for a 15-min imaging period following the incubation time ($*p = 0.0349$; $**p = 0.0010$; one-way ANOVA). **k** Percentage of microglia surrounding the lesion (within a 75 μm radius of the lesion) with processes reaching the lesion within 20 min ($*p = 0.0154$; $**p = 0.0041$; one-way ANOVA). **l** Quantification over time of the change in fluorescence intensity of the area around the lesion (not including the lesion) ($*p < 0.05$, for aglycemia vs aglycemia+EGCG, two-way ANOVA). Data are represented as mean ± SEM. Source data are provided as a Source Data file. See also Supplementary Fig. 5.

reliance to glutaminolysis in order to maintain their immune surveillance behavior for prolonged periods.

Clinical hypoglycemic events may occur from insulin overdose, hepatic or renal disease, chronic alcoholism, or in cases of hypoglycorrhachia associated with infections. Once blood glucose falls below 2.5 mM, cognitive disabilities become apparent as confusion and lethargy, and if hypoglycemia persists, symptoms can proceed to coma, seizures, and possibly permanent neuronal damage. Interestingly, during insulin-induced hypoglycemia in healthy humans, the cerebral metabolic rate of glucose consumption is decreased to a greater extent than the rate of oxygen utilization, suggesting the metabolism of an alternative carbon

source in these conditions[69]. Microdialysis measurements within the rat striatum during insulin-induced hypoglycemia revealed that glutamine concentrations matched that of sham injected rats until 60 min after hypoglycemia, after which time glutamine began to decline[70]. Magnetic resonance spectroscopy measurements in humans revealed a decrease in glutamate in healthy control and type 1 diabetes (T1D) patients exposed to 30 min of insulin-induced hypoglycemia. This is likely due to the increased metabolism of the glutamate and glutamine pools in the absence of glucose[71]. Therefore, in clinical cases of hypoglycemia, it is likely that the glutamate and glutamine pools are metabolized by glial cells to maintain TCA cycle function. This fits well with our

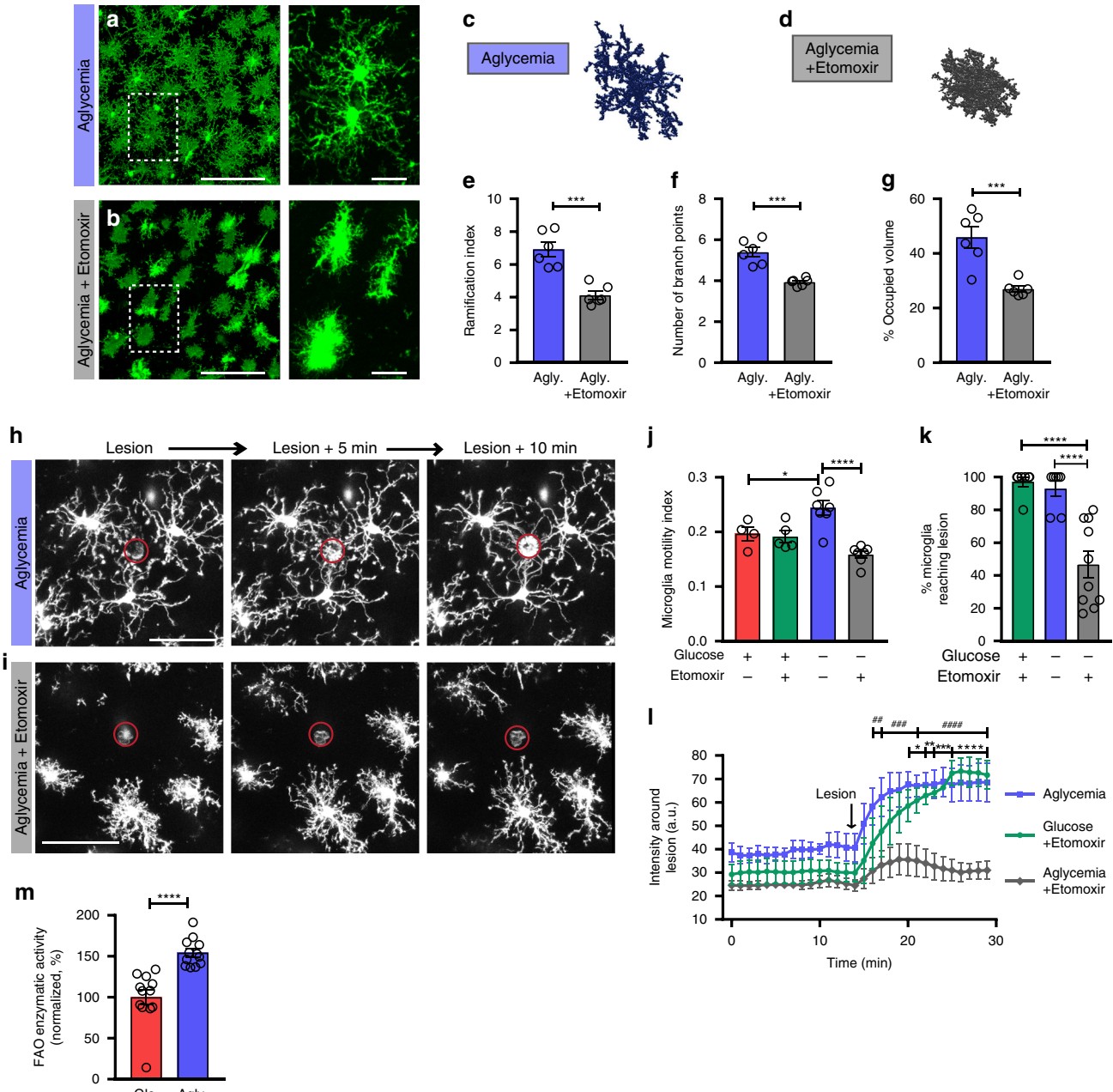

**Fig. 8 Inhibition of fatty acid oxidation reduces microglial surveillance during aglycemia in situ. a**, **b**, In acute brain slices, microglial morphology following a 60-min incubation in aglycemic aCSF **a** or aglycemic aCSF with Etomoxir **b** (for panel **a**–**g**, aglycemia: $n = 6$ mice, 12 slices, 233 cells; aglycemia+ Etomoxir: $n = 6$ mice, 11 slices, 231 cells). **c**, **d** 3DMorph rendering of representative microglial cells following a 60-min incubation in aglycemic aCSF **c** or aglycemic aCSF with Etomoxir **d**. **e** Ramification index of microglia 60 min after aglycemia or aglycemia+etomoxir incubation (***$p = 0.0002$, unpaired, two-tailed $t$-test). **f** Number of branch points per microglia (***$p = 0.0002$, unpaired, two-tailed $t$-test). **g** Percentage of brain volume occupied by microglia (***$p = 0.0010$, unpaired, two-tailed $t$-test). **h**, **i** Microglial response to a laser-induced lesion following a 60-min incubation in aglycemic aCSF **h** or aglycemic aCSF with Etomoxir **i**. **j** Microglial motility index after 60-min incubations in the indicated conditions. Shown here is the mean microglial motility for a 15-min imaging period following the incubation time (*$p = 0.0430$; ****$p = 0.000038$; one-way ANOVA; glucose: $n = 4$; glucose+Etomoxir: $n = 5$; aglycemia: $n = 7$; aglycemia+Etomoxir: $n = 7$). **k** Percentage of microglia surrounding the lesion (within a 75 µm radius of the lesion) with processes reaching the lesion within 20 min (glucose+Etomoxir vs aglycemia+Etomoxir: ****$p = 0.000021$; aglycemia vs aglycemia+Etomoxir: ****$p = 0.000074$; one-way ANOVA; glucose+Etomoxir: $n = 7$; aglycemia: $n = 7$; aglycemia+Etomoxir: $n = 10$). **l** Quantification over time of the change in fluorescence intensity of the area around the lesion (not including the lesion) (*$p < 0.05$; **$p < 0.01$; ***$p < 0.001$; ****$p < 0.0001$ for glucose+Etomoxir vs aglycemia+Etomoxir; ##$p < 0.01$; ###$p < 0.001$; ####$p < 0.0001$ for aglycemia vs aglycemia+Etomoxir, two-way ANOVA). **m** Fatty acid oxidation enzymatic activity (oxidation of ocatnoyl-CoA) of brain slices incubated in glucose-containing aCSF or aglycemic aCSF for one hour (****$p = 0.00003$ by unpaired, two-tailed $t$-test; $n = 3$ mice, 12 slices). Data are represented as mean ± SEM. Source data are provided as a Source Data file. See also supplementary Figs. 6 and 7.

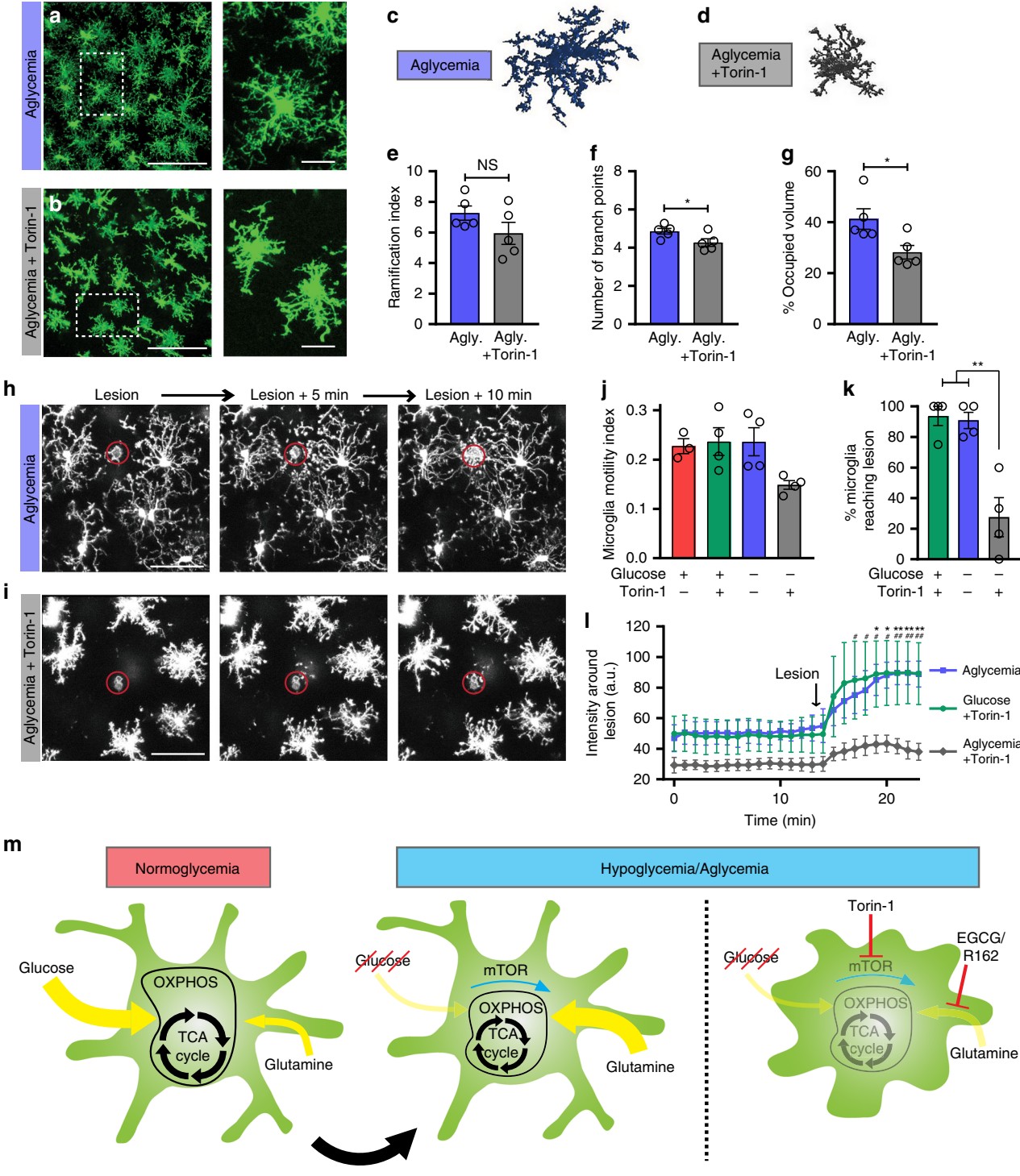

observations of glutamine as an alternative energy source for microglia, and with our measured rundown of endogenous glutamine in brain slices lacking glucose.

The early, acute phase of such metabolic pathologies represents a critical window for therapeutic prevention of functional loss. It is therefore imperative that we understand how brain cells communicate and function in that particular time frame. The observation that microglia perform critical surveillance functions even after the loss of brain energetic homeostasis therefore represents a key factor in acute neuroprotection.

In normoglycemic conditions, microglia can sense neuronal synaptic communication through diffusible signals such as glutamate and ATP[5,7,8,72,73]. In turn, microglia respond by extending their processes, contacting synapses and releasing factors like BDNF, TNFα, IL-1β or reactive oxygen species (ROS) to modulate neuronal activity[6,74–81]. In hypoglycemia or aglycemia, rapid neuronal dysregulation leads to a loss in synaptic activity, however concomitant glutamate excitotoxicity and dysregulated ATP release are likely to participate in driving the maintained, or even heightened surveillance motility of microglia observed here.

**Fig. 9 Microglial metabolic flexibility is mTOR-dependent. a, b** In acute brain slices, microglial morphology following a 60-min incubation in aglycemic aCSF (**a**) or aglycemic aCSF with Torin-1 (**b**) (for panel **a–g**, aglycemia: $n = 5$ mice, 9 slices, 228 cells; aglycemia+Torin-1: $n = 5$ mice, 11 slices, 232 cells). **c, d** 3DMorph rendering of representative microglial cells following a 60-min incubation in aglycemic aCSF **c** or aglycemic aCSF with Torin-1 **d**. **e** Ramification index of microglia 60 min after aglycemia or aglycemia+Torin-1 incubation (NS, $p > 0.05$, unpaired, two-tailed $t$-test). **f** Number of branch points per microglia (*$p = 0.0442$, unpaired, two-tailed $t$-test). **g** Percentage of brain volume occupied by microglia (*$p = 0.0264$, unpaired, two-tailed $t$-test). **h, i** Microglial response to a laser-induced lesion following a 60-min incubation in aglycemic aCSF (**h**) or aglycemic aCSF with Torin-1 (**i**) (for **h–l**, $n = 3$ slices for control, 4 slices for treatment conditions). **j** Microglial motility index after 60-min incubations in the indicated conditions. Shown here is the mean microglial motility for a 15-min imaging period following the incubation time. **k** Percentage of microglia surrounding the lesion (within a 75 μm radius of the lesion) with processes reaching the lesion within 20 min (**$p = 0.0012$ and 0.0017; one-way ANOVA). **l** Quantification over time of the change in fluorescence intensity of the area around the lesion (not including the lesion) (*$p < 0.05$; **$p < 0.01$ for glucose+Torin-1 vs aglycemia+Torin-1; #$p < 0.05$; ##$p < 0.01$ for aglycemia vs aglycemia+Torin-1, two-way ANOVA). **m** Graphical summary of the proposed microglial metabolic flexibility. Data are represented as mean ± SEM. Source data are provided as a Source Data file. See also supplementary Fig. 8.

The microglial ability to sense local damage and phagocytose even in the complete absence of glucose may be critical for process extension towards dying neurons or for repairing vascular damage during metabolic stress[1,2,82]. The microglial maintenance of mitochondrial function in aglycemia may also affect how microglia communicate with neurons, as mitochondrial ROS generation has specifically been shown to contribute to mature IL-1β production via NLRP3 inflammasome regulation[83,84].

We demonstrate that these sustained microglial functions require an intracellular metabolic pathway switch. Under baseline conditions, NAD(P)H lifetime shows that microglia are glycolytic, without excluding the possibility that glutamine is also used, as our experiments suggest using glutaminolysis blockers or changing glucose and glutamine in defined media. When glucose is absent, microglia switch to reliance on glutaminolysis, where glutamine is taken up and processed to α-ketoglutarate to enter the TCA cycle. In the brain, glutamate is rapidly cleared from the extracellular milieu, while glutamine concentration is relatively high, estimated at 385 μM[57]. Microglial glutamine uptake has been shown to occur through glutamine transporter SNAT1, and its overexpression in Rett syndrome models induces an increase in mitochondrial oxygen consumption[85,86]. Along with the lack of glutamate transporter expression in microglia, these factors indicate that microglia directly take up glutamine for energy use. The release of glutamine in the extracellular space following glutamate conversion depends on astrocytes. Therefore an interesting question remains if and how astrocytes can perform this function in the absence of glucose. Our data suggest that astrocytes may temporarily rely on fatty acid β-oxidation to maintain their conversion of glutamate to glutamine, thereby feeding microglia. In astrocytes, a complex balance between the use of glutamate for conversion to glutamine versus its direct use for energy metabolism exists, implying flexibility in the metabolic substrate use of astrocytes[87,88]. Further investigations of this possible astrocyte metabolic flexibility during acute aglycemia may uncover a new metabolic aspect of the classic glutamate-glutamine cycle[39]. It is interesting to speculate on the possibility that, just as astrocytic glycogen stores may be used to feed lactate to neurons under metabolic stress, astrocytes may likewise take up excess glutamate to convert it to glutamine and feed microglia.

Rapid adaptation to alternative metabolic fuels has been reported for immune cells of the periphery, notably activated T cells metabolize glutamine while memory T cells and macrophages can use fatty acid oxidation for metabolic support[21,22,24]. A similar switch has also been extensively studied in cancerous cells that become 'glutamine-addicted'[40]. A common aspect of this metabolic reprogramming is the central role of mTOR, which we show is also involved in microglial flexibility. As the core component of the functionally distinct multiprotein complexes mTORC1 and mTORC2, mTOR is thought to function as a master regulator of metabolism and cell proliferation. It integrates signals of nutrient availability and cellular energy status to regulate protein and lipid synthesis, autophagy, and metabolic flux[42,43]. mTORC1 signaling can stimulate transcription of mitochondrial genes through 4E-BP inhibition to support an increase in OXPHOS, increase glycolysis via HIF-1α transcription, and increase glutaminolysis by decreasing transcription of SIRT4, an inhibitor of GDH[89–91]. Another mTOR-containing complex, mTORC2 acts primarily through phosphorylation of protein kinases A, G and C (PKA, PKG, PKC) as well as Akt/PKB and PI3 kinase (PI3K), potentially eliciting faster effects[92,93]. In addition, mTORC2 and its binding partner RICTOR promote cytoskeletal rearrangements and migration by activating Rho GTPases and PKC[94–98] that could potentially contribute to microglial motility[99,100]. A recent proteomic analysis has shown that decreased levels of the microglial mTORC2-RICTOR pathway components correlate with differential metabolic programs observed in primary microglia derived from young vs old animals[101]. Adding to the complexity of mTOR signaling is that mTOR itself is regulated by glutamine or by glutaminolysis and concomitant α-ketoglutarate production[67,68,102]. This suggests a crosstalk in which mTOR in microglia participates in sensing the availability of energy resources and in adapting the metabolic pathway, and that feedback is provided by the use of glutamine.

Our results therefore demonstrate that microglial metabolic flexibility and reprogramming represents a critical feature of the brain's response to changes in glucose availability. Some aspects of the microglial metabolic signature certainly recapitulate that of peripheral immune cells, but the unique metabolic conditions encountered in the brain parenchyma establishes a brain-specific form of metabolic flexibility. It is now believed that impaired energy metabolism and altered cerebrovascular mechanisms accompany, and possibly directly contribute to most neurodegenerative diseases[103–105]. Whether in chronic brain disorders or in acute metabolic stress, a better understanding of the specific energy requirements of microglia will have key clinical implications for neuronal function and brain health.

## Methods

**Animals**. Mouse housing and experimental procedures were performed in accordance with Canadian Council on Animal Care (CCAC) regulations, with protocols approved by the University of British Columbia committee on animal care. Animals were group housed and fed ad libitum. Wild type C57BL/6 (WT) or CX3CR1$^{EGFP/+}$ (Jackson Lab strain 005582 crossed with wild type C57BL/6) mice were housed on a 12 h light/day cycle. Male and female adult mice (45-120 days old) were used. Two-week-old Sprague-Dawley rats were used for primary microglia cultures.

**Open cranial window surgery and in vivo two-photon imaging**. A round cranial window (4 mm diameter) was drilled under general anesthesia (fentanyl, 0.05 mg/kg; midazolam, 5 mg/kg; medetomidine, 0.50 mg/kg) as previously described[13]. The mice were kept under general anesthesia for the entire imaging period and were directly transferred to the microscope after the completion of the surgery. Body temperature was controlled and kept steady at 37 °C. The craniotomy was enclosed

by a custom-made head fixation ring which allowed for the head fixation of the mouse in the two-photon microscopy setup. In vivo imaging was performed using a custom-built, fully motorized, two-photon microscope equipped with a Coherent Chameleon Ultra II laser and a Zeiss W Plan-Apochromat 40×/numerical aperture 1.0 objective. The microscope is motorized and controlled by a Sutter MP285 via ScanImage (version 3.8). Microglia were imaged using a 900 nm excitation wavelength and were detected via non-descanned detectors and an ET525/50m-2P emission filter (Chroma Technology). 40–50 μm-thick z-stacks were collected using ScanImage at 1024 × 1024 pixels at a depth of approximately 100–200 μm below the cortical surface. Laser power was kept constant for each experiment and did not exceed 45 mW.

**Acute slice preparation and maintenance.** Mice were anesthetized with 3% isoflurane and decapitated. The brains were quickly removed and sliced in an ice-cold solution containing the following (in mM): 120 N-methyl-d-glucamine, 2.5 KCl, 25 NaHCO₃, 1 CaCl₂, 7 MgCl₂, 1.2 NaH₂PO₄, 20 D-glucose, 2.4 sodium pyruvate, and 1.3 sodium l-ascorbate, which was constantly oxygenated with 95% O₂ and 5% CO₂. Brains were sliced horizontally (300 μm-thick) using a vibratome (Leica VT1200S) with disposable razor blades (Canemco & Marivac). Hippocampal slices were immediately transferred to artificial cerebral spinal fluid (aCSF), which was continuously oxygenated with 95% O₂ and 5% CO₂. aCSF contained (in mM): 126 NaCl, 2.5 KCl, 26 NaHCO₃, 2 CaCl₂, 2 MgCl₂, 1.25 NaH₂PO₄, and 10 D-glucose, pH 7.3–7.4, osmolarity 300 mOsm. In aglycemic conditions, glucose was replaced with 10 mM sucrose to maintain osmolarity. Slices were allowed to recover in aCSF at 32 °C for a minimum of 30 min before imaging for time-lapse experiments, or were then incubated for one hour in specified experimental conditions before imaging or SNAPSHOT fixation[106]. All microglial dynamic imaging was performed at 32 °C and NAD(P)H FLIM experiments were performed at RT.

**Acute slice in situ two-photon imaging.** A two-photon laser scanning microscope (Zeiss LSM 7MP) with a Zeiss 20×W/1.0 NA objective coupled to a Coherent Chameleon Ultra II laser was used to image live and fixed hippocampal brain slices. Microglia were imaged in the stratum radiatum of the CA1 region at 150 ± 25 μm below the surface of the slice. At that depth, microglia show no signs of activation for up to 4 h[11,107,108]. Images for time-lapse analysis were collected at 512 × 512 pixels or 1024 × 1024 pixels using 8- or 16-line averaging. Live time series were imaged as stacks of 30 μm depth with a step size of 2 μm in the z-axis. Fixed images were acquired as stacks of 30-80 μm depth with a step size of 1–2 μm in the z-axis. EGFP was excited at 920 nm and the emission was detected with a photo-multiplier tube after passing through a 490–550 nm emission filter (Chroma, ET520/60 m). Lesions were induced by exposure of high laser power illumination to a restricted area. Inhibitors were applied at concentrations ~10–1,000 times their respective IC50 to account for brain slice (at 150 μm depth) permeability, diffusion and/or breakdown.

**Brain and slice fixation.** Mice were anesthetized with a combination of fentanyl 0.05 mg/kg, midazolam 5 mg/kg and medetomidine 0.50 mg/kg and transcardially perfused with ice-cold PBS, followed by ice-cold 4% PFA. Brains were further post-fixed by immersion in 4% PFA overnight. 300 μm-thick slices were then cut using a vibratome before imaging. For post-incubation fixed imaging experiments, brain slices were fixed following the indicated treatments by immersion in 4% PFA at 80 °C for 2 min and rinsed in 0.1 M PBS. Before imaging the slices were mounted on specially made microscope slides (Fisher Scientific) and imaged as described in the slice imaging section.

**Blood and CSF glucose measurements.** For insulin-induced hypoglycemia experiments, insulin (5 U/kg, Humulin R, Lilly) or sham (saline) was injected intraperitoneally and blood glucose was assessed every 15 min. Throughout these glucose measurements and imaging experiments, mice were anesthetized with a combination of fentanyl 0.05 mg/kg, midazolam 5 mg/kg and medetomidine 0.50 mg/kg. Using a 30-gauge needle, a drop of blood was drawn from the tail and glucose concentration was measured using a One Touch Ultra 2 blood monitoring system and One Touch Ultra test strips (LifeScan, Switzerland). CSF was collected as previously described[109]. 45–60 min after either sham or insulin (5 U/kg) injections for glucose concentration measurements. Briefly, anesthetized mice were secured in a surgical adapter and an incision through the skin and muscle layers covering the back of the neck was made to expose the base of the skull. The area over the cisterna magna was cleaned and using a sharpened glass capillary, the cisterna magna was punctured and 15–30 μl of CSF (clear and free of blood contamination) was drawn. For experiments involving CSF or aCSF glucose measurements (Fig. 2b, Supplementary Fig. 2), glucose concentrations of blood and CSF were measured using a colorimetric assay according to manufacturer's guidelines (Sigma, GAHK20).

**Two-photon NAD(P)H fluorescence lifetime imaging.** To image NAD(P)H lifetime, microglia in WT slices were identified by a 45-min incubation with DyLight 594 tomato lectin. CX3CR1$^{+/EGFP}$ mice could not be used, as it was recently confirmed that GFP expression causes an artifact in NAD(P)H measurements[54]. During imaging, slices were submerged in oxygenated aCSF (at 3 mL/min perfusion speed) in an imaging chamber. Tissue was excited at 750 nm, and emitted light was split using a 480 nm long pass dichroic mirror. Blue NAD(P)H fluorescence passed through a 460/50 m filter and was detected by a GaAsP hybrid detector (HPM-100-40 hybrid PMT, Becker and Hickl). Longer wavelengths were again split by a 575 nm long pass dichroic mirror. Green fluorescence (non-NAD (P)H autofluorescence) was collected after passing through a 535/50 m filter to be excluded from lifetime analysis, while red (DyLight 594 tomato lectin) passed through a 630/75 m filter before detection. All mirrors and filters were purchased from Chroma tech, Bellows Falls, VT. Images were taken in the stratum radiatum of CA1 hippocampus at a depth between 50 μm–80 μm below the surface of the brain slice. Images were acquired at 256 × 256 (zoom factor 10; 42.51 × 42.51 μm xy scale) over 30 s to ensure a sufficient number of photons were collected for curve fitting.

**NAD(P)H fluorescence lifetime data analysis.** NAD(P)H photons were collected by the hybrid-PMT, and detected by a TCSPC module (SPC-150, Becker and Hickl, Berlin, Germany) and SPCM software (Becker and Hickl, Berlin, Germany). Laser pulse clock information was sent to the SPC-150 module to enable lifetime calculations. Photon counts were passed to SPC Image, where decay curves for each pixel were calculated using a two-component exponential (representing free and bound NAD(P)H) by the following equation:

$$F(t) = \alpha_1 e^{-t/\tau_1} + \alpha_2 e^{-t/\tau_2} \tag{1}$$

Where $\alpha$ is the amplitude and $\tau$ is the lifetime. The data was processed with unfixed $\tau_1$ and $\tau_2$ lifetimes, which correspond to the free and bound forms of NAD(P)H, respectively. The mean lifetime of each pixel is calculated by:

$$\tau_m = \alpha_1 \tau_1 + \alpha_2 \tau_2 \tag{2}$$

Where $\alpha_1 + \alpha_2 = 1$. A spatial bin factor of 3 was used to attain a photon count >10 at the tail of the curve. A mask around the microglial soma was manually drawn, using the DyLight 594 tomato lectin fluorescence image as a guide, and avoiding neuropil signal contamination at cell edges. The average mean lifetime within a microglial mask was recorded.

**Primary microglia culture and Seahorse assays.** For primary microglial cultures, cortex from embryonic day 18 Sprague Dawley rats were obtained and gently triturated in DMEM culture media containing 10% FBS and 100 units/mL penicillin/streptomycin. Cells were filtered through 45 μm culture inserts, seeded into 100 mm culture dishes, and incubated at 37 °C and 5% CO₂ for two weeks. The mixed glia cultures were shaken (Southwest Science SBT300 Digital Orbital Shaker) with a speed of 80 r.p.m. for 4 h and the detached microglia were plated into a Seahorse 96-well plate at a density of 30 K cells/well and left to adhere overnight. Immediately following experimental media application, plates were loaded into the 96-well Seahorse XFe machine. Treatments consisted of full media (10 mM glucose + 4 mM L-glutamine), glucose only (10 mM glucose + 4 mM sucrose), glutamine only (4 mM L-glutamine + 10 mM sucrose), or deficient (14 mM sucrose for osmotic balance). An initial baseline reading was set for a total of four hours to measure the microglial metabolic response to media conditions. After four hours, a mitochondrial stress test was performed, including the sequential injection of Oligomycin (final concentration: 2 μM), FCCP (final concentration: 2 μM), and Rotenone/Antimycin A (final concentration: 2.5 μM each). Oligomycin inhibits ATP synthase, revealing the amount of OCR used by cells to fuel ATP production. FCCP is a protonophore, which disrupts the inner mitochondrial membrane and thereby dissipates the electrochemical proton gradient. In an attempt to restore the gradient, components of the ETC work at maximum rate, and the resulting OCR corresponds to the maximal respiratory capacity of the cells. Finally, Rotenone and Antimycin A inhibit complex I and III of the ETC, respectively. This completely blocks electron flow and any remaining OCR measured after this point is non-mitochondrial respiration, and was subtracted from all values. All OCR data was normalized to the initial baseline reading to control for variability in cell plating and a minimum of 6 wells were run per condition in each experiment. The experiment was repeated in three separate plates. Normalized baseline metabolic rates are considered as the average of the last three readings before metabolic drug injection (after a four-hour incubation with treatment media), and maximum respiration is considered as the third measurement following FCCP application.

**MTT assay.** Primary microglia were cultured with DMEM/F12 media with 10% FBS and 1% penicillin/streptomycin in a 96-well plate at 20 K cells/well. Prior to MTT assays, media was carefully aspirated and microglia were washed with deficient media (0 mM glucose, 0 mM glutamine). Experimental media with defined carbon source (10 mM glucose + 4 mM glutamine; 10 mM glucose + 0 mM glutamine; 0 mM glucose + 4 mM glutamine; 0 mM glucose + 0 mM glutamine) without FBS was added for a 4-h incubation, followed by addition of freshly prepared MTT solution. After a 20-min incubation, MTT solution was replaced by DMSO, and the plate was agitated for 2 min. DMSO extract (80 μl) was transferred to a fresh 96-well plate and OD measured at 540 nm.

**Glutamine assay.** Glutamine level was measured from mouse brain slices using a glutamine colorimetric assay kit (Biovision, CA). The brain slices were recovered

for 30 min in a chamber with aCSF aerated with 95% $O_2$ / 5% $CO_2$ at 32 °C. Some slices were collected immediately (time 0) and some slices were transferred to a separate chamber with different conditions for 1-h incubation. The assay is based on the hydrolysis of glutamine to glutamate, which produces a stable signal for the colorimetric measurement of tissue glutamine. The signal is proportional to the concentration of glutamine in the brain slices. The background signal without hydrolysis enzyme mix was subtracted to exclude the contamination of glutamate in the tissue. As recommended by the manufacturer, the samples were deproteinized using 10 K cutoff spin column. Glutamine levels were normalized to protein concentration of the samples. The optical density was measured at 450 nm using a plate reader.

**Fatty acid oxidation assay.** Fatty acid oxidation was measured from mouse brain slices using a fatty acid oxidation assay kit (Biomedical research service center, University of Buffalo, NY). The brain slices were recovered for 30 min in a chamber with aCSF aerated with 95% $O_2$/5% $CO_2$ at 32 °C. The brain slices were transferred to normoglycemic (10 mM glucose) and aglycemic (10 mM sucrose) conditions and incubated for 1 h before collection for the analysis. The fatty acid beta oxidation activity was measured by oxidation of the substrate octanoyl CoA, which is coupled to NADH-dependent reduction of INT to formazan. The signal is proportional to the activity of fatty acid oxidation in the samples. The optical density was measured at 492 nm.

**Reagents.** EGCG (Cayman Chemical Company 70935, 100 μM), Torin-1 (Tocris 4247, 10 μM), R162 (EMD Millipore 538098, 200 μM), Etomoxir (Cayman Chemical Company 11969, 100 μM), Perhexiline (Tocris 5166, 200 μM).

**Image processing and analysis.** The morphology of microglia from fixed slices images were quantified using the published 3DMorph script[55]. Briefly, this generated data for microglial ramification index (extent of the microglial vs microglia size), number of branch points (following skeletonization of the microglia) and % occupied volume (territorial volume occupied by microglia as measured by a convex polygon around the cell and its processes, expressed as a percentage of total volume of the image). For motility index quantification, microglia were imaged every minute, z-stacks were aligned and projected stack images were binarized. The number of new pixels (additions) at each time point was normalized to the total number of pixels in the cells (cell size), therefore giving a quantification of motility normalized to cell size. For lesion experiments, fluorescence intensity around lesion was obtained by measuring the mean fluorescence of an ROI around the lesion over time. The ROI was drawn so as not to include the lesion itself as it generates autofluorescence. The percentage of microglia responding to the lesion was calculated as the % of microglia that extend processes and reach the lesion within a 20-min imaging period, among the microglia that have their cell body within 75 μm of the lesion. Some low magnification images of microglia are shown as 3D renderings obtained with Zen (Zeiss). All other images are projected z-stacks obtained using ImageJ (NIH).

**Statistical analysis.** All values shown in the graphs are the mean ± standard error of mean. For each experiment, replicates are described in the figure legends. Statistical tests used are described for each analysis in the corresponding figure legends. The exact $p$-values are indicated in the figure legends.

**Reporting summary.** Further information on experimental design is available in the Nature Research Reporting Summary linked to this paper.

## Data availability

The data that support the findings of this study are available from the corresponding author upon reasonable request. The source data underlying Figs. 1d, 2b–f, i, k, l, 3e–g, i, j, m, n, 4d–f, 5a–d, 6c, d, 7e–g, j–l, 8e–g, j–m, 9e–g, j–l, and supplementary Figs. 1b, c, 2a, b, 3a–d, 4, 5c, f–h, 6c–h, 7a–c; 8c–e are provided as a Source Data file.

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

## Acknowledgements

This work was supported by grants from the Fondation Leducq and Canadian Institute for Health Research (FDN-148397, B.A.M.). L.P.B. was supported by CIHR Banting Fellowship, Heart and Stroke Foundation, Michael Smith Foundation for Health Research and Fonds de Recherche Santé Québec. E.M.Y. was supported by CIHR Canada Graduate Scholarship—Doctorate. A.K. was supported NSERC Canada Graduate Scholarship—Masters.

## Author contributions

L.P.B., E.M.Y. and B.A.M. conceived and designed the study. L.P.B. performed microglial imaging experiments. E.M.Y. performed NAD(P)H imaging and Seahorse experiments. N.L.W. performed NAD(P)H imaging. A.K. performed microglial culture assays. H.B.C. performed biochemical and culture assays. L.P.B., E.M.Y., A.K., H.B.C and N.L.W. contributed data and analysis. L.P.B. and E.M.Y. wrote the manuscript. All authors contributed to the editing of the manuscript.

## Competing interests

The authors declare no competing interests.
