## [Peer Review File · Nature Communications]

Reviewers' comments:

Reviewer #1 (Remarks to the Author):

Bernier and colleagues present a set of compelling data to suggest that microglia are capable of metabolizing glutamine in hypoglycemic conditions. The work will be of broad interest to immunologists and neuroscientists.

The paper is very well written and the experiments well conducted.

We believe that two modifications will significantly improve the strength of the presentation.

1) Determination whether astrocytes are capable of metabolizing glutamine. We feel the paper would be strengthened if it were determined that astrocytes do or do not consume glutamine in conditions of stress. This experiment will add to our understanding of the glial reaction to hypoglycemia and provide insight whether this mechanism is specific to microglia among brain cells.

2) The cell culture data add little to the publication and should be deleted. This step is required because of the well known and crippling limitations of cultured microglial-like cells including specifically primary microglia and the line used by these investigators (for which the report of their generation by 'spontaneous immortalization' includes nothing more in the way of characterization than Iba1 and CD68 staining both wretched in quality). If mTOR dependence is to be established, then an in vivo system should be used. The optimal experiment could use slice cultures with tissue taken from floxed mTOR mutants crossed to CX3CR1-cre or other microglial driver.

Minor points:

1) Figures should generally not be referenced in the introduction.

Reviewer #2 (Remarks to the Author):

This paper reports the interesting observation that, unlike most other brain cell types, brain microglia can continue to carry out their surveying functions even when glucose is removed from the medium, as a result of an ability to switch to glutamine (Gln) as a substrate for TCA cycle operation. This is reminiscent of metabolic switching performed by other immune cells and cancer cells, and will be of interest to a wide range of neuroscientists investigating microglial function in conditions of metabolic compromise.

The paper is potentially acceptable, but needs some control experiments added, and some confusion of nomenclature sorting out.

Main comments

(1) The authors use fluorescence lifetime imaging of NAD(P)H to try to infer the relative rates of glycolysis and oxidative phosphorylation (oxphos) in microglia. This raises some questions:

Since NADH cannot be distinguished from NADPH, please state the likely relative concentrations of these molecules when describing the approach on page 5 (see Ref 44 for help on this point).

As Fig 2 of ref 44 (Yaseen et al.) makes clear, the possible locations of NADH are more complex than the schematic in Fig 1a of the current paper. The authors would do well to acknowledge that fact, and mention (and show in Fig 1a) that NADH is not just free or bound to oxphos enzymes, but can also be bound to glycolytic and TCA cycle enzymes.

The schematic in Fig 1 of this paper suggests that the authors believe that a low mean lifetime reflects the presence of a high level of free NADH generated by glycolysis; this is consistent with their statement (page 6, top) that the short lifetime implies a high ratio of glycolysis to oxphos in microglia. However, the difference in NADH lifetime between microglia and neuropil in Fig 1d is much larger than that seen in the paper cited (ref 44) when oxidative phosphorylation is inhibited. Furthermore, Ref 44 shows that NADH fluorescence lifetime is reduced (never increased) when blocking either glycolysis or oxphosph, raising the question of how the authors can claim that measuring differences of lifetime allows them to conclude that (page 6, top) "microglia have a comparatively higher ratio of glycolysis to oxphos"; surely, on the authors' ideas, blocking glycolysis while maintaining oxphos should prolong the lifetime? This should be explained. How can the authors interpret a reduced lifetime as meaning less oxphos as opposed to less glycolysis (based on ref 44)?

To ensure that the authors interpretation of their data is correct (i.e. short lifetime = more glycolysis; long lifetime = more oxphosph) for the case of microglia, they should add data showing the microglial NADH lifetime change in response to

(i) block of oxphosph (e.g. with antimycin),

(ii) block of glycolysis with iodoacetate while maintaining the TCA cycle and oxphosph by superfusing pyruvate (they already have data doing this with Gln providing a TCA substrate but it is of interest to know what happens if the more normal pyruvate is used).

These experiments should be just a few days work, but they may reveal that the causes of changes in NAD(P)H lifetime are more complex than the authors admit, which would require some rewriting.

(2) The lack of morphology change in aglycemia

This may in part reflect the weakness of the insult. Morrison & Filosa (2013, J Neuroinflammation) found larger ramification changes. This paper should be cited and the data in the current paper should be compared with the Morrison & Filosa data.

(3) Lines 130-134.

Why does the neuropil NAD(P)H lifetime decrease with time in brain slices? It would be good to mention, in connection with the acute slice experiments, that no glutamine was added to the medium, so any use of Gln that occurs in aglycemia is utilising endogenous Gln.

In line 134 "bound NADH" is mentioned as though there were only one bound form. Is it the case that NADH bound to TCA cycle enzymes, to glycolytic enzymes and to the electron transport chain have the same lifetime? This should be discussed, e.g. in connection with the discussion of Fig 2 of ref 44 mentioned above.

Removing glucose will inhibit both glycolysis and oxphos and so it is hard to predict whether the lifetime should increase or decrease. It will presumably only increase if there is an alternative substrate to sustain the TCA cycle such as Gln. This should be discussed at the point (line 133) where the "important" increase of lifetime is mentioned for when glucose is removed.

(4) Cultured microglia

Cultured microglia almost certainly have properties different from those of microglia in situ. Do the SIM-A9 cells recapitulate the results of Figs 1-4 for microglia in situ?

(5) Block of "glutaminolysis".

Line 155. EGCG does not block glutaminolysis, which is mediated by the enzyme glutaminase. It

blocks glutamate dehydrogenase, the downstream enzyme that converts glutamate made by glutaminase into alpha-ketoglutarate which enters the TCA cycle (see <https://cancerres.aacrjournals.org/content/69/20/7986>). This should be made clear (here and on lines 191, 203 and 219). How specific is EGCG for blocking GDH? Why not block glutaminase activity to check it gives the same result?

It is stated that EGCG significantly blocked MTT oxidoreduction. To claim this, it is necessary to add a p value on the graph comparing the 2 blue bars in Fig 5a. Is this significant after correction for the multiple corrections in this figure?

(6) Fig 5c

Why is there O₂ consumption even with no glucose or Gln present?
Should "control" in the key be written as "glucose+Gln" for clarity?
Help the mito-illiterate reader (on the figure or in the text (bottom of page 9)) by explaining exactly what oligomycin, CCCP, rotenone/AA do.

(7) Fig. 6

Line 191. The authors need to explain why NAD(P)H lifetime decreases on removing glucose; although this is consistent with ref 44 as noted above, and presumably reflects a change in the balance of free and bound NADH in various locations, the authors' text will prime readers to think that reducing glycolysis (by removing glucose) will increase lifetime. The reader needs to be reminded that removing glucose inhibits the TCA cycle and oxphos as well.

(8) Fig 7

It is very surprising that microglial ramification persists after an hour of complete aglycemia when ion gradients may be expected to run down dramatically. Was there any substrate for the TCA cycle included in the superfusion solution for these experiments (e.g. Gln)?

In Fig 7j is the motility larger in the absence of glucose (compare the blue and red bars)?

In Fig 7l is there an obvious explanation for the different initial fluorescence levels?

(9) Discussion

Although the authors' discovery about the role of Gln is interesting, in reality is there ever a situation in which glucose is absent from the extracellular solution when Gln is present (perhaps insulin overdose?) or do they tend to vary together? Microdialysis data in the literature might provide an answer.

Other points

(1) Page 4 line 43. Glutamine is NOT the deaminated form of glutamate (it's the other way round - Gln is Glu plus an NH₂ group)! Alpha-ketoglutarate is a deaminated form of glutamate.

(2) Line 93, page 6: Although it is in the figure, maybe state what concentration glucose decreases from.

(3) Line 123: Was neuronal activity raised or lowered in ref 49 (don't make the reader have to go and look up that essential fact)?

(4) Fig 8a. Add the unnormalized values to the figure legend.

Reviewer #3 (Remarks to the Author):

In the manuscript, entitled "Microglia metabolic flexibility supports immune surveillance of the brain parenchyma", Bernier and York et al investigated energy substrates that could support microglia's immune surveillance role. Using an insulin-induced hypoglycemia model, the authors first found that there was no change in microglia motility or morphology after laser-induced injury in insulin treated mice compared to shams. Next they used a series of in vitro assays to demonstrate that microglia can maintain oxidative phosphorylation when glutamine was the only carbon source. This was also evidenced by an in situ measurement of NAD(P)H levels under concurrent aglycemia and EGCG treatment. In addition, the authors found that blockade of the GDH enzyme by EGCG during aglycemia was able to induce a ramified morphology in microglia with significantly less branch points and occupancy of space. Importantly, the authors demonstrated that these ramified microglia had less motility compared to those under aglycemia only treatment, and were deficient in responding to laser-induced lesions. Finally, they showed that a mTOR inhibitor Torin-1 was able to induce a ramified microglial morphology similar to that from EGCG treatment, which was also associated with deficits in motility and response to laser-induced lesions. Based on these results, Bernier and York et al concluded that in the absence of glucose, the switch of microglial metabolism from glycolysis to glutaminolysis is dependent on mTOR, and this flexibility is important for microglia's role in immune surveillance. This study investigated an important issue that can potentially increase our understanding of the metabolic mechanisms of microglia's immune surveillance roles under certain conditions. The manuscript is well written, the technique of using two-photon imaging to measure NAD(P)H levels is novel, the results are clear. However, there are fundamental flaws in the experimental design that render the results and conclusions less convincing.

Major concerns

Fatty acid metabolism is important for the reactivity of microglia^{1,2}. However, the current study failed to take fatty acids into consideration. First, in Figure 5, experiment media was deficient media without FBS, the main source of fatty acids for cells in culture. Therefore, is conducted in a fatty acid-free condition. The results shown in Figure 5 are likely the combined effect of glucose, glutamine and fatty acids deficiency. Second, EGCG inhibits fatty acid synthase^{3,4}, limiting the availability of fatty acids to microglia can in turn inhibit the oxidative phosphorylation of fatty acids. The authors didn't mention this point, and there is no detail of EGCG treatment length for in situ experiments shown in Figures 6 and 7. Prolonged EGCG treatment might result in depletion of available fatty acids to microglia, and yielded similar results as Figure 5. Therefore, based on the current data, it is impossible to rule out the contribution of fatty acids in microglial metabolism and function when glucose is absent.

Minor points

In the introduction, page 4 second paragraph, "Glutamine is the deaminated form of glutamate, ..." Glutamine and glutamate should be switched in this sentence.

Figure 5a, the authors claimed " MTT metabolism ..., but was significantly decreased in the glucose-only or deficient conditions", but there is no statistical analysis to support this claim.

Reference

1. PMID: 31238131
2. PMID: 31318452
3. PMID: 11239828

Reviewers' comments:

Reviewer #1 (Remarks to the Author):

Bernier and colleagues present a set of compelling data to suggest that microglia are capable of metabolizing glutamine in hypoglycemic conditions. The work will be of broad interest to immunologists and neuroscientists.

The paper is very well written and the experiments well conducted.

We believe that two modifications will significantly improve the strength of the presentation.

1) Determination whether astrocytes are capable of metabolizing glutamine. We feel the paper would be strengthened if it were determined that astrocytes do or do not consume glutamine in conditions of stress. This experiment will add to our understanding of the glial reaction to hypoglycemia and provide insight whether this mechanism is specific to microglia among brain cells.

We have now included significant additions to the manuscript which reveal an important contribution of astrocyte-microglia communication in maintaining immune surveillance in the brain. We show compelling data suggesting astrocytes are also metabolically flexible and in that way maintain their ability to convert glutamate into glutamine, which then feeds microglial glutaminolysis. Metabolic flexibility of astrocytes and their use of glutamate or glutamine directly for energy has been suggested previously (See works from Mckenna. *Front. Endocrinol.* 2013. PMID: 24379804 or *Neurochem Res.* 2012. PMID 23079895). Astrocytic expression of CPT1, an enzyme necessary for fatty acid oxidation also points to their use of fatty acids. We have included in the manuscript data showing that specific inhibition of CPT1 (by inhibitors etomoxir or perhexiline) induces microglial deramification and reduced motility, likely by decreasing the astrocytic conversion of glutamate to glutamine. We also show that fatty acid oxidation (FAO) enzymatic activity (of brain tissue) increases in response to aglycemia, suggesting an astrocytic increase in FAO to support glutamate-glutamine conversion.

To determine the cell specificity (astrocyte or microglia) of this FAO increase, we have also evaluated the specific FAO activity of both microglia and astrocytes in cell cultures. However, we have not included it in the manuscript due to concerns raised by two reviewers regarding the use of cell culture. We provide the results here instead and we could include it in the manuscript upon request. As shown in the graph below, astrocytes significantly increase their FAO enzymatic capacity when faced with the absence of glucose and glutamine.

We have also added several sections to the discussion related to the role of astrocytes in terms of providing glutamine to microglia. We have also added text and references (McKenna 2012; Schousboe et al. 2014) related to previous investigations on astrocytic metabolic flexibility and use of glutamate for metabolism.

We have also added sections discussing the expression of CPT1, a key enzyme for fatty acid oxidation (see comments from reviewer 3). CPT1 was shown in 2017 by Jernberg et al (J. Neurochem. 142:407, ref 62 and line 295 to line 301 in manuscript) to be exclusively expressed in astrocytes and our results suggest that in the absence of glucose, astrocytes can use fatty acid oxidation to maintain their production of glutamine and thereby “feed” microglia to maintain microglial immune surveillance.

In general, we believe that astrocytes are key in maintaining the metabolic environment in the brain and as such, likely play a critical role in supporting the availability of nutrients that microglia may use for immune surveillance. However, this study is aimed at evaluating the metabolic responses of immune cells of the brain and comparing microglial metabolic flexibility to that of peripheral immune cells in the unique context of brain metabolism. Therefore, a full study of astrocytic metabolism and possible metabolic flexibility seems beyond the scope of our work. We could, upon request, provide data from astrocytes in culture (MTT assay in defined-carbon media), keeping in mind the aforementioned concerns of primary culture use.

2) The cell culture data add little to the publication and should be deleted. This step is required because of the well known and crippling limitations of cultured microglial-like cells including specifically primary microglia and the line used by these investigators (for which the report of their generation by 'spontaneous immortalization' includes nothing more in the way of characterization than Iba1 and CD68 staining both wretched in quality). If mTOR dependence is to be established, then an in vivo system should be used. The optimal experiment could use slice cultures with tissue taken from floxed mTOR mutants crossed to CX3CR1-cre or other microglial driver.

The reviewers comment on the limitations of cell culture to interpret microglial function is well received. In the revised manuscript, we have taken out all SIM-A9 microglial cell line data. However, we do believe in the usefulness of cell cultures in isolating the activities of particular cell types and cell autonomous functions without possible confounding factors from other neighbouring cell types. For that reason, we have repeated the key experiment of MTT oxidoreduction in defined-carbon media in primary microglia, as opposed to SIM-A9, and included that in the new manuscript. It is certainly true that primary microglia are not exactly like *in vivo* microglia, but they are still more representative than SIM-A9 cell lines and we believe they support and complement our data, which is still largely collected in *in vivo* or *in situ* paradigms.

Using brain slice cultures instead of the acute brain slice model we use in this manuscript would, similarly to cell cultures, likely artificially affect the metabolic state of microglia. Microglia in acute brain slices are less likely to have long-term transcriptional changes driven by the experimental condition affect their metabolic state. Organotypic brain slices are cultured for days and are therefore less likely to represent the original baseline steady-state metabolic signature of microglia at the onset of the treatment or experiment. While we agree that floxed mutants may be useful, the breeding of a new mouse line is a significant undertaking and would be better suited for an in-depth study of the subcellular mechanistic pathways underlying the regulation of metabolic flexibility.

Minor points:

1) Figures should generally not be referenced in the introduction.

The figure reference has been removed from the introduction and is now cited later in the text and is now part of figure 9.

Reviewer #2 (Remarks to the Author):

This paper reports the interesting observation that, unlike most other brain cell types, brain microglia can continue to carry out their surveying functions even when glucose is removed from the medium, as a result of an ability to switch to glutamine (Gln) as a substrate for TCA cycle operation. This is reminiscent of metabolic switching performed by other immune cells and cancer cells, and will be of interest to a wide range of neuroscientists investigating microglial function in conditions of metabolic compromise.

The paper is potentially acceptable, but needs some control experiments added, and some confusion of nomenclature sorting out.

Main comments

(1) The authors use fluorescence lifetime imaging of NAD(P)H to try to infer the relative rates of glycolysis and oxidative phosphorylation (oxphos) in microglia. This raises some questions:

Since NADH cannot be distinguished from NADPH, please state the likely relative concentrations of these molecules when describing the approach on page 5 (see Ref 44 for help on this point).

The revised manuscript now includes more detailed descriptions on the possible contribution of NADPH to the overall NAD(P)H signal measured.

Page 6, line 90: “Furthermore, since NADH fluorescence cannot be experimentally distinguished from the closely related species NADPH 51, we hereafter refer to our measurements as NAD(P)H. However, NADPH likely represents a negligible number of photons in our NAD(P)H measurements, as previous studies quantifying pyridine dinucleotides in the brain report NADH concentrations to be approximately 5 to 10 fold higher than NADPH 52,53. Accordingly, our readings of NAD(P)H lifetimes in response to metabolic manipulations (iodoacetate or antimycin A, Supplementary fig.1) perform as expected from the NADH species. Thus, our data can be interpreted within the simplified schematic where short NAD(P)H lifetime correlates with glycolysis and longer NAD(P)H lifetime with OXPHOS activity.”

Since we believe our manuscript should be focused on microglial metabolism and its importance in acute microglial responses to metabolic challenges, we tried to keep this methodological aspect of NADH FLIM measurements minimal. However, to potentially address this question regarding NADPH contribution, we have performed preliminary experiments with physcion, a 6-phosphogluconate dehydrogenase inhibitor (Lin. Nat Cell Biol. 2015. PMID: 26479318). This decreases NADPH production and therefore reduces the NADPH contribution to the acquired NAD(P)H signal. Preliminary data (shown below) shows no significant change in the acquired endogenous NAD(P)H photon counts and therefore suggests NADPH is negligible in our NAD(P)H measurements. Also shown is a control assay (NADPH ELISA) to confirm the drug actually decreases NADPH content.

As Fig 2 of ref 44 (Yaseen et al.) makes clear, the possible locations of NADH are more complex than the schematic in Fig 1a of the current paper. The authors would do well to acknowledge that fact, and mention (and show in Fig 1a) that NADH is not just free or bound to oxphos enzymes, but can also be bound to glycolytic and TCA cycle enzymes.

We agree that possible NADH binding partners are complex. Accordingly, we have now included an updated version of the schematic diagram in Fig. 1a to now include potential other sources of free and

bound NADH. Upon this reviewer's advice, we have also added two critical control experiments (now Supplementary figure 1, described in more details in another comment below) using glycolytic blocker iodoacetate and ETC inhibitor Antimycin A. We have also included additional text in the results section to discuss this issue.

Page 5, Line 80: "Multiple studies have reported a decrease in mean NADH lifetimes when the ETC is inhibited 47,48, suggesting that the mitochondrial NADH signal is mainly correlated with Complex I binding and OXPPOS activity 49,50, although additional minor binding partners exist, such as lactate dehydrogenase and malate dehydrogenase. Following ETC inhibition with antimycin A treatment (Complex III inhibitor, Supplementary fig. 1), we observed a decrease in the mean lifetime of microglia, supporting the model that bound, long lifetime NADH reflects ETC activity. In addition, glycolytic inhibition with iodoacetate (GAPDH inhibitor, applied along with pyruvate to maintain TCA cycle and ETC activity (Supplementary fig. 1) induced an increase in mean lifetime in microglia, confirming that glycolytic activity correlates with measurements of free, short lifetime NADH."

And a later sentence: "Thus, our data can be interpreted within the simplified schematic where short NAD(P)H lifetime correlates with glycolysis and longer NAD(P)H lifetime with OXPPOS activity. "

The schematic in Fig 1 of this paper suggests that the authors believe that a low mean lifetime reflects the presence of a high level of free NADH generated by glycolysis; this is consistent with their statement (page 6, top) that the short lifetime implies a high ratio of glycolysis to oxphos in microglia. However, the difference in NADH lifetime between microglia and neuropil in Fig 1d is much larger than that seen in the paper cited (ref 44) when oxidative phosphorylation is inhibited.

Based on the model presented above (and the references cited therein), it is expected that OXPPOS will contribute to the bound (long lifetime) component of NAD(P)H while glycolysis will produce free (short lifetime) NAD(P)H. Therefore, given that the resting state NAD(P)H mean lifetime is much lower in microglia than in neuropil, it follows that there must be a higher ratio of free:bound NAD(P)H in microglia. The measurement performed in Figure 1 is comparing the lifetime between two cell populations in control conditions, rather than in response to OXPPOS inhibition. It is possible that differences in enzymatic expression and activity allow microglia to maintain a lower mean lifetime than that observed after OXPPOS inhibition across cortical tissue in Yaseen et al. Importantly, this is a difficult comparison to make, as in Figure 1, we are measuring baseline states in two cell types, rather than a response to pharmacological manipulation averaged across tissue mainly composed of neurons (as in Yaseen et al.).

Furthermore, Ref 44 shows that NADH fluorescence lifetime is reduced (never increased) when blocking either glycolysis or oxphosph, raising the question of how the authors can claim that measuring differences of lifetime allows them to conclude that (page 6, top) "microglia have a comparatively higher ratio of glycolysis to oxphos"; surely, on the authors' ideas, blocking glycolysis while maintaining oxphos should prolong the lifetime? This should be explained. How can the authors interpret a reduced lifetime as meaning less oxphos as opposed to less glycolysis (based on ref 44)?

As noted above, at page 6 (top, in the initial manuscript; now page 6 line 102), we are comparing microglia to neuropil and no manipulation of glycolysis or OXPHOS are done at that point. Therefore the shorter lifetime of microglia at rest can be interpreted as microglia having a more “glycolytic” profile than cells of the neuropil.

To address the issue of a decreasing lifetime obtained by blocking either glycolysis or OXPHOS in Yaseen et al., we believe this makes sense in the context of our model. If OXPHOS is inhibited, there will be less bound NAD(P)H, and therefore a decrease in the mean lifetime. The effect of blocking glycolysis is slightly less intuitive, but makes sense if one considers a more-or-less direct relationship between NADH reduction and oxidation. If fewer NAD⁺ molecules are reduced (by blocking glycolysis), this will decrease the free NADH component. However, this will also result in fewer NADH molecules being available to bind to Complex I, resulting in a concomitant decrease in the bound NAD(P)H pool. Therefore, it is only possible for the mean lifetime to increase if the initial reduction of NAD⁺ to NADH is decreased (ie. glycolysis is inhibited) at the same time as the bound component is maintained (ie TCA and ETC metabolism from alternate carbon sources). Indeed, we do observe an increased mean lifetime in microglia under aglycemic conditions, which we suggest is the result of glutamine metabolism to maintain TCA cycle and subsequent OXPHOS, and further supports our model. Importantly, the measurements taken in Yaseen et al are representative of the entire field of view of rat cortex, and the signal is therefore likely dominated by the neuronal metabolic signal. This result then supports our findings, as we observe that neuropil mean lifetime does not increase in aglycemic conditions. We predict that neurons are less metabolically flexible than glia, which are able to fuel their TCA and ETC metabolism with alternate carbon sources.

To directly compare our results with those of Yaseen et al, we also performed experiments with antimycin A and Iodoacetate, as proposed by the reviewer (see next comment below). In these experiments, the microglial NAD(P)H lifetime increased upon glycolysis blockade (IAA with pyruvate, see answer below for more details on microglia). However, the lifetime signal in the neuropil decreased, similar to what Yaseen et al. observed. This further highlights the differences in metabolic flexibility between cells of the neuropil and microglia. The results pertaining to microglia are included in the manuscript (Supplementary figure 1), but the neuropil results are shown below (IAA: iodoacetate, pyr: pyruvate, AA: antimycin A).

To ensure that the authors interpretation of their data is correct (i.e. short lifetime = more glycolysis; long lifetime = more oxphosph) for the case of microglia, they should add data showing the microglial NADH lifetime change in response to

(i) block of oxphosph (e.g. with antimycin),

(ii) block of glycolysis with iodoacetate while maintaining the TCA cycle and oxphosph by superfusing pyruvate (they already have data doing this with Gln providing a TCA substrate but it is of interest to know what happens if the more normal pyruvate is used).

These experiments should be just a few days work, but they may reveal that the causes of changes in NAD(P)H lifetime are more complex than the authors admit, which would require some rewriting.

We appreciate the suggestion of this experiment, and agree that it is an important test of metabolic perturbations on NAD(P)H FLIM measurements. In the revised manuscript, we have included these experiments as Supplementary Figure 1. Notably, antimycin A treatment induced an expected decrease in the mean NAD(P)H lifetime, and iodoacetate combined with pyruvate induced an increase (although not statistically significant, $p=0.06$) in the mean NAD(P)H lifetime of microglia. This fits our interpretation that more glycolysis (antimycin inhibition of OXPHOS) means shorter lifetime and that more OXPHOS (inhibition of glycolysis with IAA while maintaining TCA and OXPHOS with pyruvate) means longer lifetime.

We have also incorporated the following text in the manuscript (page 5, line 83): “Following ETC inhibition with antimycin A treatment (Complex III inhibitor, Supplementary fig. 1), we observed a decrease in the mean lifetime of microglia, supporting the model that bound, long lifetime NADH reflects ETC activity. In addition, glycolytic inhibition with iodoacetate (GAPDH inhibitor, applied along with pyruvate to maintain TCA cycle and ETC activity (Supplementary fig. 1) induced an increase in mean lifetime in microglia, confirming that glycolytic activity correlates with measurements of free, short lifetime NADH.”

(2) The lack of morphology change in aglycemia

This may in part reflect the weakness of the insult. Morrison & Filosa (2013, J Neuroinflammation) found larger ramification changes. This paper should be cited and the data in the current paper should be compared with the Morrison & Filosa data.

It is likely that the large morphological changes seen in Morrison & Filosa are a reflection of lack of oxygen (by MCAO-induced ischemic stroke). This paradigm not only causes metabolic restriction, but is also a traumatic insult that will induce other cellular damage signaling cascades that will result in a microglial morphological change. It is expected that if we paired our aglycemic treatment with anoxia, we would observe a much more dramatic morphological response by microglia. However, in this investigation, we were concerned with the inherent metabolic flexibility of microglia rather than their response to the pathological insult of OGD, which certainly results in activation of immune signaling pathways in addition to metabolic changes. The reference for Morrison & Filosa 2013 is now added to the stroke section in the introduction of the revised manuscript (Ref 32).

(3) Lines 130-134.

Why does the neuropil NAD(P)H lifetime decrease with time in brain slices? It would be good to mention, in connection with the acute slice experiments, that no glutamine was added to the medium, so any use of Gln that occurs in aglycemia is utilising endogenous Gln.

No exogenous glutamine was added to the aCSF in any of our brain slice experiments. We have made that clearer in this section: “we imaged acute hippocampal slices of CX3CR1EGFP/+ mice during complete aglycemia (0 mM glucose in artificial CSF (aCSF) - absence of glucose in the tissue confirmed experimentally in Supplementary fig. 2. No exogenous carbon sources were added)” Regarding the glutamine content in brain slices, we have also performed experiments to confirm that glutamine is present in our brain slice experiments. These data are now included and described in the text (page 8, line 135).

Briefly, these experiments confirmed two points. 1. Glutamine is still present in appreciable concentrations after the one-hour incubation we used throughout the study. The glutamine content is still 47.5% of the baseline value at time 0 (immediately after slice cutting). It therefore confirms that glutamine is available to be used by microglia in the paradigms employed (this is shown in Supplementary figure 2). 2. Although some of this decrease is likely due to dilution in the large volume of aCSF, some of the decrease is also likely due to glutaminolysis via the incorporation of glutamine into the TCA cycle (through α -ketoglutarate) and therefore measured as reduced glutamine content in the slice. Accordingly, the decrease was more prominent (to 33.8%) in aglycemic conditions, suggesting more glutamine is being used (as an alternate carbon source).

We also performed similar glutamine content experiments as a control for EGCG. Since EGCG blocks the conversion of glutamine into α -ketoglutarate, this leads to less glutamine being metabolized by cells. Therefore, an increase in glutamine (albeit not significant) content is observed relative to a one-hour incubation in control aCSF. We have not included that in the manuscript but it is shown in the graph below.

Regarding the point raised about neuropil NAD(P)H lifetime decreasing over time, we found this to be a consistent and interesting result in and of itself. While we felt that investigations confirming the cause of this decrease would detract from the focus on microglial metabolism, we do hypothesize that the

observed decrease is a result of reduced mitochondrial activity in neurons with time after slicing. Importantly, the baseline metabolic profile of microglia remained stable throughout the duration of all experiments.

In line 134 “bound NADH” is mentioned as though there were only one bound form. Is it the case that NADH bound to TCA cycle enzymes, to glycolytic enzymes and to the electron transport chain have the same lifetime? This should be discussed, e.g. in connection with the discussion of Fig 2 of ref 44 mentioned above.

As mentioned in the previous comment, we agree with the reviewer on the more complex nature of NADH binding and have adapted the schematic diagram and text accordingly to downplay our simplified view of NADH binding. As a general interpretation, we believe the bound form mainly represents binding to ETC components and OXPHOS activity, but this is likely not responsible for 100% of the signal, and we have adapted the text to reflect that. As noted in the earlier comment, we have also included a discussion of the finer details of this interpretation (in the introduction of the NAD(P)H-FLIM technique in the results section) so that the reader is aware of this potential limitation.

Removing glucose will inhibit both glycolysis and oxfhos and so it is hard to predict whether the lifetime should increase or decrease. It will presumably only increase if there is an alternative substrate to sustain the TCA cycle such as Gln. This should be discussed at the point (line 133) where the “important” increase of lifetime is mentioned for when glucose is removed.

As pointed out by the reviewer, the increase observed can only be expected if an alternate source of carbon sustains the TCA cycle. Our data suggests glutamine is this alternate carbon source and we have now clarified the text to reflect this interpretation.

Page 9, line 159: “Therefore, microglia show a unique ability to sustain oxidative phosphorylation in the absence of glucose, suggesting that they are metabolizing an alternate carbon source to supply TCA metabolites and maintain OXPHOS.”

The subsequent section goes on to explore glutamine as a possibility for this ‘alternate carbon’ source.

This increase in lifetime was also observed in new experiments performed on the advice of the reviewer, where IAA combined with pyruvate (as the alternate source to ultimately sustain the TCA cycle) led to an increased mean lifetime in microglia. This is now shown in Supplementary figure 1.

(4) Cultured microglia

Cultured microglia almost certainly have properties different from those of microglia *in situ*. Do the SIM-A9 cells recapitulate the results of Figs 1-4 for microglia *in situ*?

This point was also raised by another reviewer. Generally, the results with SIM-A9 recapitulate the data obtained *in situ* as glutamine seems to be sufficient for microglia viability and function, and metabolic flexibility is observed. But we agree that cultured microglia are very different from *in situ* microglia. Due to the concerns on SIM-A9 use, in the revised manuscript, we have taken out all SIM-A9 microglial cell

line data. However, we do believe in the usefulness of cell cultures in isolating the activities of particular cell types and cell autonomous functions without possible confounding factors from other neighbouring cell types. For that reason, we have repeated the key experiment of MTT oxidoreduction in defined-carbon media in primary microglia, as opposed to SIM-A9 cell lines, and included that in the new manuscript. It is certainly true that primary microglia are not exactly like *in vivo* microglia, but they are still more representative than SIM-A9 cell lines and we believe they support and complement our data, which is still primarily collected in *in vivo* or *in situ* paradigms. Of particular importance is the use of Seahorse Metabolic Flux Analyzer, a tool widely used for metabolic analysis in high-impact publications which provides important replication and support of our metabolic interpretations from *in situ* NAD(P)H lifetime imaging.

(5) Block of “glutaminolysis”.

Line 155. EGCG does not block glutaminolysis, which is mediated by the enzyme glutaminase. It blocks glutamate dehydrogenase, the downstream enzyme that converts glutamate made by glutaminase into alpha-ketoglutarate which enters the TCA cycle (see <https://cancerres.aacrjournals.org/content/69/20/7986>). This should be made clear (here and on lines 191, 203 and 219). How specific is EGCG for blocking GDH? Why not block glutaminase activity to check it gives the same result?

EGCG does indeed block glutamate dehydrogenase (GDH) and not glutaminase. We have used the term glutaminolysis in the broad sense of “use of glutamine as a metabolic fuel”, and not in the strict sense of glutamine catabolism. Due to the importance of the glutamate-glutamine cycle in the brain between astrocytes and neurons, it was important for us not to prevent the interconversion of these molecules by inhibiting glutaminase, as it would cause large off-target effects in the tissue not directly related to metabolism. To address the metabolism of glutamate specifically, EGCG was chosen, as it inhibits the conversion of glutamate to α -ketoglutarate. This point has been clarified in the text:

Added to page 12, line 227: “Importantly, EGCG will block the conversion of glutamate to α -ketoglutarate, thereby inhibiting its metabolism without directly preventing the glutamate-glutamine cycle needed for neuronal function in brain tissue.”

Also, as mentioned in an above comment, we have performed new control experiments that confirm that EGCG affects glutamine metabolism in the expected manner (see experiments on glutamine content in brain slices treated with or without EGCG mentioned above).

It is stated that EGCG significantly blocked MTT oxidoreduction. To claim this, it is necessary to add a p value on the graph comparing the 2 blue bars in Fig 5a. Is this significant after correction for the multiple corrections in this figure?

As significant concerns have been raised around using the SIM-A9 cultures, these data have been removed from the manuscript. Nevertheless, the reviewer is correct, we can not directly compare these two groups. The statement should have been that in conditions where EGCG is present with only

glutamine as a carbon source, we observed a significant decrease in MTT oxidoreduction compared to the glucose + glutamine group. In the absence of EGCG, no such decrease is observed. The adjusted graph is provided here (but not in the manuscript anymore).

(6) Fig 5c

Why is there O2 consumption even with no glucose or Gln present?

The basal level of OCR present in deficient media likely reflects trace amounts of metabolites present in the Seahorse base media, which may include amino acids, or possibly the autolysis of these microglial cells in extreme nutrient starvation. Additionally, it could reflect ROS production rather than metabolism, which would also be measured as a 'consumption' of oxygen in the media. This would also explain the relatively small change in OCR in these wells upon addition of the ATP synthase inhibitor, Oligomycin, which is used to reveal the ATP-linked respiration of cells.

Should "control" in the key be written as "glucose+Gln" for clarity?

This has been corrected in this figure, as well as in Supplementary figure 4.

Help the mito-illiterate reader (on the figure or in the text (bottom of page 9)) by explaining exactly what oligomycin, CCCP, rotenone/AA do.

The following text has been added to the description of the Seahorse Mito Stress test:

Page 11, line 203: "After the 4-hour incubation with specific carbon sources, we ran a mitochondrial stress test, consisting of successive applications of the metabolic inhibitors oligomycin (to inhibit ATP synthase and reveal ATP-linked respiration), carbonyl cyanide-4 (trifluoromethoxy) phenylhydrazone (FCCP; a mitochondrial protonophore driving maximal respiration), and rotenone with antimycin A (inhibitors of complex I and complex III, respectively, to completely block ETC activity and reveal any remaining non-mitochondrial oxygen consumption)."

(7) Fig. 6

Line 191. The authors need to explain why NAD(P)H lifetime decreases on removing glucose; although this is consistent with ref 44 as noted above, and presumably reflects a change in the balance of free and bound NADH in various locations, the authors' text will prime readers to think that reducing glycolysis (by removing glucose) will increase lifetime. The reader needs to be reminded that removing glucose inhibits the TCA cycle and oxphos as well.

In this particular experiment, the NAD(P)H lifetime decreases in these conditions as we are preventing both glycolysis, and glutaminolysis with EGCG. Therefore, as noted by the reviewer, TCA and OXPHOS metabolism are inhibited also (since the alternate use of glutamine as a carbon source for the TCA cycle is now prevented by EGCG-GDH blockade). This is compared with the aglycemia conditions (shown in the left bar of figure 6c, but also in more details in figure 3) without EGCG inhibition of glutaminolysis, in which glutamine acts as alternate carbon source to replenish TCA and OXPHOS metabolism, and is therefore associated with an increased NAD(P)H lifetime.

The following text was added in the manuscript for clarification (page 12, line 231):

“In these conditions, a decrease in the NAD(P)H lifetime is expected, as aglycemia without the ability to metabolize alternative carbon substrates and replenish the TCA cycle and ETC metabolism will result in a corresponding decrease of both free and bound NAD(P)H.”

(8) Fig 7

It is very surprising that microglial ramification persists after an hour of complete aglycemia when ion gradients may be expected to run down dramatically. Was there any substrate for the TCA cycle included in the superfusion solution for these experiments (e.g. Gln)?

No exogenous glutamine was added to the aCSF in any of our brain slice experiments. As detailed in the previous findings, the microglia adapt and maintain their metabolism (and therefore likely also their own ion gradients) in these conditions by metabolizing endogenous glutamine as an alternate carbon source to maintain their morphology, motility and damage-sensing functions. This has been clarified in the text (page 13, line 249): “We subjected brain slices from CX3CR1^{EGFP/+} mice to one hour of complete aglycemia without addition of an alternate exogenous carbon source, and microglia still showed typical ramified morphology as quantified in Fig.3. (Fig. 7a).”

In Fig 7j is the motility larger in the absence of glucose (compare the blue and red bars)?

Yes, the microglial motility in the absence of glucose is increased relative to control conditions. This has been updated on the graph and figure legend. This same parameter was also significantly increased in figure 3j.

In Fig 7l is there an obvious explanation for the different initial fluorescence levels?

Yes, the initial fluorescence levels are higher in aglycemia conditions as a result of the slightly longer (more ramified) microglial processes in these conditions. Therefore, there is no difference in the fluorescence itself, but rather more nearby GFP-positive processes around the ROI prior to laser-induced lesion. As seen in figures 3e,g, aglycemia induced small increases (although not statistically significant) in ramification and volume occupied. This observation is also visible on figure 3m and the newly added figure 8l. Since this is not statistically significant, we did not discuss it specifically in the manuscript but we could do so upon request.

(9) Discussion

Although the authors' discovery about the role of Gln is interesting, in reality is there ever a situation in which glucose is absent from the extracellular solution when Gln is present (perhaps insulin overdose?) or do they tend to vary together? Microdialysis data in the literature might provide an answer.

Hypoglycemic events may occur from insulin overdose, hepatic or renal disease, chronic alcoholism, or in cases of hypoglycorrhachia associated with infections. Once blood glucose falls below 2.5 mM, cognitive disabilities become apparent as confusion and lethargy, and if hypoglycemia persists, symptoms can proceed to coma, seizures, and possibly permanent neuronal damage. Interestingly, during insulin-induced hypoglycemia in healthy humans, the cerebral metabolic rate of glucose consumption is decreased to a greater extent than the rate of oxygen utilization, suggesting the metabolism of an alternative carbon source in these conditions. [Lubow et al., *Am J Physiol Endocrinol Metab.* 2006. PMID: 16144821]

Microdialysis measurements within the rat striatum during insulin-induced hypoglycemia revealed that glutamine concentrations matched that of sham injected rats until 60 minutes after hypoglycemia, after which time glutamine began to decline (Silverstein. *Ann Neurol.* 1990. PMID: 1979220). Magnetic resonance spectroscopy measurements in humans revealed a decrease in glutamate in healthy control and type 1 diabetes (T1D) patients exposed to 30 minutes of insulin-induced hypoglycemia. This is likely due to the increased metabolism of the glutamate and glutamine pools in the absence of glucose. [Terpstra. *J Cereb Blood Flow Metab.* 2014. PMID: 24549182]. Therefore, in clinical cases of hypoglycemia, it is likely that the glutamate and glutamine pools are metabolized by glial cells to maintain TCA cycle function. This fits well with our observations of glutamine as an alternative energy source for microglia, and with our measured rundown of endogenous glutamine in brain slices lacking glucose.

Other points

(1) Page 4 line 43. Glutamine is NOT the deaminated form of glutamate (it's the other way round - Gln is Glu plus an NH₂ group)! Alpha-ketoglutarate is a deaminated form of glutamate.

We thank the reviewer for noticing this error – it has been corrected:

“Glutamate, the deaminated form of glutamine, is the primary excitatory neurotransmitter in the CNS.”

(2) Line 93, page 6: Although it is in the figure, maybe state what concentration glucose decreases from.

The following text has been added (page 7, line 112):

“Moderate to severe hypoglycemia was achieved 45-60 minutes following injection, with blood and CSF glucose concentration decreasing to an average of 3.27 mM and 3.21 mM, respectively (compared with 22.55 mM and 7.19 mM 45-60 minutes after vehicle control injection; Fig. 2b).”

(3) Line 123: Was neuronal activity raised or lowered in ref 49 (don't make the reader have to go and look up that essential fact)?

Activity was lowered. This has been changed in the text from “altered” to “lowered”.

(4) Fig 8a. Add the unnormalized values to the figure legend.

Due to concerns around SIM-A9 cultures, Figure 8a has been removed from the manuscript. This normalization was necessary to adjust for the variability between each experiment. Each experimental day had its own set of control conditions so were normalized to that specific day's control. Also, MTT assays do not allow for adjustment to protein content unlike most other cell culture biochemical assays. We would still like to provide the reviewer with the unnormalized values, which were: Vehicle full= 0.474; Vehicle Glucose= 0.363; Vehicle Glutamine= 0.443; Vehicle Deficient= 0.349; Torin-1 Full= 0.391; Torin-1 Glucose= 0.271; Torin-1 Glutamine= 0.235; Torin-1 Deficient= 0.238.

Reviewer #3 (Remarks to the Author):

In the manuscript, entitled "Microglia metabolic flexibility supports immune surveillance of the brain parenchyma", Bernier and York et al investigated energy substrates that could support microglia's immune surveillance role. Using a insulin-induced hypoglycemia model, the authors first found that there was no change in microglia motility or morphology after laser-induced injury in insulin treated mice compared to shams. Next they used a series of in vitro assays to demonstrate that microglia can maintain oxidative phosphorylation when glutamine was the only carbon source. This was also evidenced by an in situ measurement of NAD(P)H levels under concurrent aglycemia and EGCG treatment. In addition, the authors found that blockade of the GDH enzyme by EGCG during aglycemia was able to induce a ramified morphology in microglia with significantly less branch points and occupancy of space. Importantly, the authors demonstrated that these ramified microglia had less motility compared to those under aglycemia only treatment, and were deficient in responding to laser-induced lesions. Finally, they showed that a mTOR inhibitor Torin-1 was able to induce a ramified microglial morphology similar to that from EGCG treatment, which was also associated with deficits in motility and response to laser-induced lesions. Based on these results, Bernier and York et al concluded that in the absence of glucose, the switch of microglial metabolism from glycolysis to glutaminolysis is

dependent on mTOR, and this flexibility is important for microglia's role in immune surveillance. This study investigated an important issue that can potentially increase our understanding of the metabolic mechanisms of microglia's immune surveillance roles under certain conditions. The manuscript is well written, the technique of using two-photon imaging to measure NAD(P)H levels is novel, the results are clear. However, there are fundamental flaws in the experimental design that render the results and conclusions less convincing.

Major concerns

Fatty acid metabolism is important for the reactivity of microglia^{1,2}. However, the current study failed to take fatty acids into consideration. First, in Figure 5, experiment media was deficient media without FBS, the main source of fatty acids for cells in culture. Therefore, is conducted in a fatty acid-free condition. The results shown in Figure 5 are likely the combined effect of glucose, glutamine and fatty acids deficiency.

We agree with the reviewer that the potential use of fatty acids by brain cells is an interesting aspect that is worth investigating. We have therefore performed multiple experiments on that aspect that are now included in our revised manuscript and considerably improves the dataset. With these new additions, we now show data on the three main groups of metabolic substrates: carbohydrates, amino acids, and fatty acids. To investigate the possible use of fatty acids for cellular energy through β -oxidation, we tested the effect of CPT1 inhibition, the rate-limiting step in FAO with the commonly-used blocker etomoxir and another CPT1 blocker perhexiline. The results shown in Figure 8 and Supplementary figure 7 as well as in the main text (pages 14-15 and 19) suggest that FAO plays a role in the maintenance of microglial surveillance in the absence of glucose. The expression pattern of CPT1 (references are in the main text, and we have added a meta-analysis of transcription levels of *Cpt1a* from published databases in Supplementary figure 7 suggests that astrocytes are more likely than other cell types to use FAO during aglycemia, and we hypothesize that they rely on fatty acids to maintain the glutamate-glutamine cycle that then feeds glutamine to microglia. Notably, a key result from Jernberg et al. (PMID: 28512781, cited in our manuscript) shows through immunohistofluorescence that the proteins CPT1a and CPT2 are only expressed in astrocytes (figure 6 in Jernberg et al., observed in all brain regions studied in adult mice).

We have also performed several experiments testing for fatty acid oxidation enzymatic activity with a FA β -oxidation assay kit. In brain slices, aglycemia induced a significant increase in FAO activity and this is now shown in figure 8m. We also tested that in cultured primary astrocytes and microglia and observed that astrocytes, but not microglia, increase the FAO enzymatic activity upon removal of carbon sources glucose and glutamine. Since reviewers expressed concerns on the use of *in vitro* culture, these data are not included in the manuscript but could be added upon request. These results are shown below.

Additionally, publicly-available transcriptomic datasets have been used to compare the transcription level of *Cpt1a* in various brain cell types, and clearly show that this enzyme necessary for FAO is most highly transcribed in astrocytes (Supplementary figure 7).

Polyunsaturated fatty acids (PUFAs) are also known to affect microglial-driven neuroinflammation in a manner that is not directly linked to their metabolism to produce cellular energy. The actions of PUFAs on microglia are driven through long-term transcriptional changes and modulate the polarization of microglia in the context of cellular activation and neuroinflammation. This is a very interesting process that warrants deeper investigations, however this is not directly relevant to the potential use of fatty acids in acute metabolic flexibility of cells.

FBS was avoided in these assays so that glucose and glutamine metabolism could be definitively measured in our defined media. The composition in FBS is neither well defined nor well controlled between batches. It is notoriously difficult to identify components of FBS, and it may include carbohydrates, amino acids and fatty acids in various quantities. We nonetheless performed these experiments with primary microglia in glucose+glutamine, glucose-only, glutamine-only or deficient media while also supplementing with 2% FBS. The addition of FBS (2%, GIBCO) to the primary microglia before and during the treatment with the 4 conditions led to MTT oxidoreduction results very similar to those obtained without FBS (shown in Fig 5 of the revised manuscript). Namely, MTT oxidoreduction was similar between the glucose-only and the glutamine-only conditions, and glutamine-only was enough to maintain oxidoreduction at a significantly higher level than in deficient conditions. The results obtained with FBS are shown below but because of the difficulty in identifying the exact composition of FBS in terms of carbon sources, and the fact that reviewers have expressed concerns over *in vitro* culture assays, we have not included that in the manuscript. We could include it upon request.

Second, EGCG inhibits fatty acid synthase^{3,4}, limiting the availability of fatty acids to microglia can in turn inhibit the oxidative phosphorylation of fatty acids. The authors didn't mention this point, and there is no detail of EGCG treatment length for in situ experiments shown in Figures 6 and 7. Prolonged EGCG treatment might result in depletion of available fatty acids to microglia, and yielded similar results as Figure 5. Therefore, based on the current data, it is impossible to rule out the contribution of fatty acids in microglial metabolism and function when glucose is absent.

EGCG treatment was performed by bath incubation for 60 minutes in either control or aglycemic aCSF for acute brain slices experiments. This clarification was added to the text description of figure 6 and 7.

Page 12, line 225: “However, after 60 minutes in the presence of the glutaminolysis inhibitor epigallocatechin gallate (EGCG), the NAD(P)H lifetime decreased upon glucose removal, showing that glutaminolysis is necessary to maintain mitochondrial function in microglia (Fig. 6a-c).”

Page 13, line 251: “However, inhibiting glutaminolysis by blocking the key enzyme GDH with EGCG during 60 minutes of aglycemia caused a visible microglial deramification, where cell bodies became enlarged with short and stubby protrusions (Fig. 7b).”

In MTT assays on SIM-A9 microglia (Figure 5 in the previous manuscript), EGCG was present throughout the defined carbon media treatment (4 hours), however it only inhibited viability and metabolism in glutamine only conditions. This experiment is now removed from the manuscript as reviewers expressed concerns on the use of SIM-A9 cultures and *in vitro* cultures in general. Since the effect of EGCG we observed was rapid (1-hour incubation in slices, and visible within minutes in supplementary video 5), and that most references showing inhibition show an effect in the range of 2-48 hours (for example, Sagano et al. PMID 11239828 which was mentioned here by the reviewer, or Brusselmans et al. JBC. PMID 12918062), we believe any properties of EGCG via fatty acid synthase inhibition likely have minimal impact in our assays.

Furthermore, we have performed control experiments to confirm that EGCG does block the entry of glutamine into the TCA cycle. While these data are not included in the manuscript, they are shown below. As would be expected if EGCG blocks conversion of glutamine into alpha-ketoglutarate, EGCG

treatment induced an increase in the remaining glutamine content of brain slices incubated for one hour in either control or aglycemia conditions (+/- EGCG). This confirms that EGCG acts as expected as a blocker of GDH.

Minor points

In the introduction, page 4 second paragraph, "Glutamine is the deaminated form of glutamate, ..." Glutamine and glutamate should be switched in this sentence.

We thank the reviewer for noticing this error – it has been corrected:

"Glutamate, the deaminated form of glutamine, is the primary excitatory neurotransmitter in the CNS."

Figure 5a, the authors claimed " MTT metabolism ..., but was significantly decreased in the glucose-only or deficient conditions", but there is no statistical analysis to support this claim.

As concerns have been raised by reviewers around using the SIM-A9 cultures, these data have been removed from the manuscript. Nevertheless, an adjusted graph showing all statistical comparisons is provided here (but is no longer in the manuscript).

Reference

1. PMID: 31238131
2. PMID: 31318452
3. PMID: 11239828

REVIEWERS' COMMENTS:

Reviewer #1 (Remarks to the Author):

The authors have addressed reviewer concerns aptly with new data and expanded discussion as requested.

Reviewer #2 (Remarks to the Author):

The authors have reviewed this paper satisfactorily and I have no further comments.

It should be published now, as it provides significant novel information on microglial bioenergetics.

Reviewer #3 (Remarks to the Author):

the authors have adequately addressed the concerns of this reviewer.